# nextGEMS: entering the era of kilometer-scale Earth system modeling

Hans Segura[1,*], Xabier Pedruzo-Bagazgoitia[2,*], Philipp Weiss[3,*], Sebastian K. Müller[4,*], Thomas Rackow[2,*], Junhong Lee[1,*], Edgar Dolores-Tesillos[5,*], Imme Benedict[6,*], Matthias Aengenheyster[2], Razvan Aguridan[2,7], Gabriele Arduini[2], Alexander J. Baker[8], Jiawei Bao[9], Swantje Bastin[1], Eulàlia Baulenas[7], Tobias Becker[2], Sebastian Beyer[10], Hendryk Bockelmann[11], Nils Brüggemann[1], Lukas Brunner[12], Suvarchal K. Cheedela[10], Sushant Das[13], Jasper Denissen[2], Ian Dragaud[12], Piotr Dziekan[14], Madeleine Ekblom[15], Jan Frederik Engels[11], Monika Esch[1], Richard Forbes[2], Claudia Frauen[11], Lilli Freischem[3], Diego García-Maroto[16], Philipp Geier[2], Paul Gierz[10], Álvaro González-Cervera[16], Katherine Grayson[7], Matthew Griffith[2], Oliver Gutjahr[1], Helmuth Haak[1], Ioan Hadade[2], Kerstin Haslehner[17], Shabeh ul Hasson[18], Jan Hegewald[19], Lukas Kluft[1], Aleksei Koldunov[10], Nikolay Koldunov[10], Tobias Kölling[1], Shunya Koseki[20], Sergey Kosukhin[1], Josh Kousal[2], Peter Kuma[21], Arjun U. Kumar[1], Rumeng Li[22], Nicolas Maury[23], Maximilian Meindl[17], Sebastian Milinski[2], Kristian Mogensen[2], Bimochan Niraula[11], Jakub Nowak[14], Divya Sri Praturi[1], Ulrike Proske[24], Dian Putrasahan[1], René Redler[1], David Santuy[16], Domokos Sármány[2], Reiner Schnur[1], Patrick Scholz[10], Dmitry Sidorenko[10], Dorian Spät[17], Birgit Sützl[2], Daisuke Takasuka[25], Adrian Tompkins[26], Alejandro Uribe[21], Mirco Valentini[2], Menno Veerman[6], Aiko Voigt[17], Sarah Warnau[6,27], Fabian Wachsmann[11], Marta Wacławczyk[14], Nils Wedi[2], Karl-Hermann Wieners[1], Jonathan Wille[28], Marius Winkler[1], Yuting Wu[1], Florian Ziemen[11], Janos Zimmermann[11], Frida A.-M. Bender[21], Dragana Bojovic[7], Sandrine Bony[23], Simona Bordoni[4], Patrice Brehmer[29], Marcus Dengler[30], Emanuel Dutra[31], Saliou Faye[32], Erich Fischer[28], Chiel van Heerwaarden[6], Cathy Hohenegger[1], Heikki Järvinen[15], Markus Jochum[33], Thomas Jung[10], Johann H. Jungclaus[1], Noel S. Keenlyside[20], Daniel Klocke[1], Heike Konow[1], Martina Klose[22], Szymon Malinowski[14], Olivia Martius[5], Thorsten Mauritsen[21], Juan Pedro Mellado[12], Theresa Mieslinger[1], Elsa Mohino[16], Hanna Pawłowska[14], Karsten Peters-von Gehlen[11], Abdoulaye Sarré[32], Pajam Sobhani[34], Philip Stier[3], Lauri Tuppi[15], Pier Luigi Vidale[8], Irina Sandu[2], and Bjorn Stevens[1]

[1]Max Planck Institute for Meteorology, Hamburg, Germany
[2]European Centre for Medium-Range Weather Forecasts, Bonn, Germany
[3]Department of Physics, University of Oxford, Oxford, United Kingdom
[4]University of Trento, Trento, Italy
[5]Institute of Geography, University of Bern, Bern, Switzerland
[6]Meteorology and Air Quality Group, Wageningen University, Wageningen, the Netherlands
[7]Barcelona Supercomputing Center, Barcelona, Spain
[8]National Centre for Atmospheric Science and Department of Meteorology, University of Reading, Reading, United Kingdom
[9]Institute of Science and Technology Austria, Klosterneuburg, Austria
[10]Alfred Wegener Institute, Helmholtz Centre for Polar and Marine Research, Bremerhaven, Germany
[11]German Climate Computing Center, Hamburg, Germany
[12]Department of Earth System Sciences, University of Hamburg, Hamburg, Germany
[13]Department of Earth and Atmospheric Sciences, National Institute of Technology Rourkela, Rourkela, India
[14]Institute of Geophysics, Faculty of Physics, University of Warsaw, Warsaw, Poland

[15]Institute for Atmospheric and Earth System Research/Physics, University of Helsinki, Helsinki, Finland
[16]Departamento de Física de la Tierra y Astrofísica, Universidad Complutense de Madrid, Madrid, Spain
[17]Department of Meteorology and Geophysics, University of Vienna, Vienna, Austria
[18]HAREME Lab, Earth and Society Research Hub (ESRAH), University of Hamburg, Hamburg, Germany
[19]Gauß-IT-Zentrum, Braunschweig University of Technology (GITZ), Braunschweig, Germany
[20]Geophysical Institute, University of Bergen, Bergen, Norway
[21]Department of Meteorology, Stockholm University, Stockholm, Sweden
[22]Institute of Meteorology and Climate Research, Karlsruhe Institute of Technology, Karlsruhe, Germany
[23]LMD/IPSL, CNRS, Université Pierre et Marie Curie, Paris, France
[24]Hydrology and Environmental Hydraulics Group, Wageningen University, Wageningen, the Netherlands
[25]Department of Geophysics, Tohoku University, Sendai, Japan
[26]Abdus Salam International Centre for Theoretical Physics, Trieste, Italy
[27]Wetsus, European Centre of Excellence for Sustainable Water Technology, Leeuwarden, the Netherlands
[28]Institute for Atmospheric and Climate Science, ETH Zurich, Zurich, Switzerland
[29]Institut de Recherche pour le Développement, Dakar, Senegal
[30]GEOMAR Helmholtz Centre for Ocean Research Kiel, Kiel, Germany
[31]Instituto Português do Mar e da Atmosfera, Lisbon, Portugal
[32]Institut Sénégalais de Recherches Agricoles, Dakar, Senegal
[33]Niels Bohr Institute, University of Copenhagen, Copenhagen, Denmark
[34]Latest Thinking GmbH, Hamburg, Germany
[*]These authors contributed equally to this work.

**Correspondence:** Hans Segura (hans.segura@mpimet.mpg.de), Xabier Pedruzo-Bagazgoitia (xabier.pedruzo@ecmwf.int), Philipp Weiss (philipp.weiss@physics.ox.ac.uk), Sebastian K. Müller (sebastian.mueller@unitn.it), Thomas Rackow (thomas.rackow@ecmwf.int), Junhong Lee (junhong.lee@mpimet.mpg.de)

**Abstract.** The Next Generation of Earth Modeling Systems (nextGEMS) project aimed to produce multi-decadal climate simulations, for the first time, with resolved kilometer-scale (km-scale) processes in the ocean, land, and atmosphere. In only three years, nextGEMS achieved this milestone with the two km-scale Earth system models, ICOsahedral Non-hydrostatic model (ICON) and Integrated Forecasting System coupled to the Finite-volumE Sea ice-Ocean Model (IFS-FESOM). nextGEMS was based on three cornerstones: 1) developing km-scale Earth system models with small errors in the energy and water balance, 2) performing km-scale climate simulations with a throughput greater than one simulated year per day, and 3) facilitating new workflows for an efficient analysis of the large simulations with common data structures and output variables. These cornerstones shaped the timeline of nextGEMS, divided into four cycles. Each cycle marked the release of a new configuration of ICON and IFS-FESOM, which were evaluated at hackathons. The participants in hackathons included experts from climate science, software engineering, and high-performance computing, as well as users from the energy and agricultural sectors. The continuous efforts over the four cycles allowed us to produce 30-year simulations of ICON and IFS-FESOM, spanning the period 2020–2049 under the SSP3-7.0 scenario. The throughput was about 500 simulated days per day on the Levante supercomputer of the German Climate Computing Center (DKRZ). The simulations employed a horizontal grid of about 5 km resolution in the ocean and 10 km resolution in the atmosphere and land. Aside from this technical achievement, the simulations allowed us to gain new insights into the realism of ICON and IFS-FESOM. Beyond its timeframe, nextGEMS builds the

foundation of the Climate Change Adaptation Digital Twin developed in the Destination Earth initiative and paves the way for future European research on climate change.

# 1 Introduction

The advent of exascale supercomputers and progress in numerical modeling opens the door to a new way of simulating our Earth system (Schär et al., 2020; Slingo et al., 2022). Several international initiatives aim to represent kilometer-scale (km-scale) processes explicitly using horizontal grid spacings equal to or finer than 10 km globally in the atmosphere (e.g., Satoh et al., 2008), land (e.g., Kollet and Maxwell, 2006), and ocean (e.g., Maltrud and McClean, 2005). Such models or simulations are referred to as "km-scale simulations" or "km-scale models" in this manuscript. Representing km-scale processes explicitly makes it possible to simulate more accurately the horizontal and vertical transfer of mass and energy and the circulation that it entails. Naturally, the finer the horizontal grid spacing, the better km-scale processes are represented. In practice, km-scale models bring climate simulations to a level of granularity that has long been proven necessary to understand regional climate impact and support climate adaptation and mitigation.

Deep convective motions in the atmosphere, for example, redistribute moisture and energy, influencing the tropical vertical temperature profile and the global hydrological cycle (Kuang, 2010; Prein et al., 2017; Tian and Kuang, 2019; Bao et al., 2024). Km-scale atmospheric simulations, also referred to as storm-resolving simulations, represent deep convection, capturing most of the characteristics of mesoscale convective systems (Peters et al., 2019; Prein et al., 2020; Becker et al., 2021), convectively coupled equatorial waves (Miura et al., 2007; Holloway et al., 2013; Senf et al., 2018), and tropical cyclones (Gentry and Lackmann, 2010; Judt et al., 2021; Baker et al., 2024). Km-scale simulations also provide a more detailed representation of the land in terms of surface topography and heterogeneity and its impact on local-, regional-, and large-scale circulations (Sandu et al., 2019). Previous studies showed that such simulations improve the representation of atmospheric blockings (Woollings et al., 2018; Schiemann et al., 2020), weaken the soil moisture–precipitation feedback (Lee and Hohenegger, 2024), and impact the circulations generated by surface-radiation interactions such as land-sea breezes (Birch et al., 2015). Km-scale oceanic simulations, also referred to as eddy-resolving simulations, represent mesoscale eddies (Hewitt et al., 2022), increasing the global kinetic energy (Chassignet et al., 2020) and vertical transport of heat (Griffies et al., 2015). Previous studies showed that the higher resolution also impacts the internal variability of the ocean (Penduff et al., 2010), the timing of Antarctic sea ice decrease, and the magnitude of the projected sea level rise (van Westen and Dijkstra, 2021; Rackow et al., 2022).

The early insights gained from storm- and eddy-resolving simulations were a strong motivation for the Next Generation of Earth Modelling Systems (nextGEMS) project funded by the European Horizon 2020 programme. nextGEMS aimed to build the next generation of km-scale Earth system models, namely the ICOsahedral Non-hydrostatic model (ICON; Hohenegger et al., 2023) and the Integrated Forecasting System (IFS; Rackow et al., 2025b). The latter can be run together with either the Finite-VolumE Sea ice–Ocean Model (FESOM; Scholz et al., 2019), used in this project, or the Nucleus for European Modelling of the Ocean model (NEMO; Madec and the NEMO System Team, 2023).

While ICON and IFS-FESOM both represent km-scale processes by using horizontal grid spacings of 10 km or finer, their history and philosophy differ. ICON relies on a simple framework and aims to minimize the use of parameterization schemes. In the atmosphere, it parameterizes only small-scale processes related to turbulence, cloud microphysics, and radiation and does not employ any scheme for convection (Hohenegger et al., 2023). The atmospheric component of IFS-FESOM, on the other hand, operates as a weather model and incorporates years of model tuning to obtain accurate forecasts. In the atmosphere, it employs sophisticated parameterization schemes for various processes including convection (ECMWF, 2023b; Rackow et al., 2025b). The two approaches of ICON and IFS-FESOM are complementary and allow us to examine whether explicitly representing km-scale processes with horizontal grid spacings of 10 km or finer is enough to capture the main features of our climate and to what extent the simulation quality depends on the ability to fine-tune the remaining small scales.

nextGEMS had the visionary goal to produce and analyze, for the first time, multi-decadal km-scale climate simulations with coupled atmosphere, land, and ocean under a Shared Socioeconomic Pathway (SSP) scenario. To facilitate the model development, nextGEMS explored new ways to foster the collaboration of project participants with expertise in software engineering, model development, and climate physics. The timeline of nextGEMS was divided into four cycles. Each cycle marked the release of new simulations with ICON and IFS-FESOM, which were prepared for and analyzed at large hackathons with over 100 participants. Two main problems had to be solved to produce multi-decadal km-scale simulations. First, we had to improve the computational throughput and simplify the analysis of simulation data. Second, we had to achieve simulations with an energetically consistent climate, i.e., a near-stationary climate with a top-of-atmosphere (TOA) energy balance close to zero with global conservation of mass and energy and no near-surface temperature drift.

nextGEMS achieved this visionary goal in only three years. In Cycle 4, we produced km-scale simulations of our climate system over multiple decades with a competitive throughput of about 500 simulated days per day (SDPD). Here, we document these simulations and review the evolution of both models, ICON and IFS-FESOM. Throughout the project, we gained knowledge of technical and scientific aspects and encountered positive and negative surprises when analyzing the simulation data. To create a storyline, we translated that learning process into four questions:

1. Radiation balance (Section 4.1): Can we adjust the radiative properties and formation mechanisms of clouds in order to correct the global radiation balance and limit the drifts in the global surface temperature?

2. Key features of mean climate (Section 4.2): Are key features of the mean climate correctly represented in a km-scale simulation with an energetically consistent climate?

3. Local- to synoptic-scale phenomena (Section 4.3): Does an energetically consistent climate constrain the patterns of local-, meso- and synoptic-scale phenomena?

4. Time scales of regional patterns (Section 4.4): Over what simulation times do regional patterns emerge?

While climate simulations with a horizontal grid spacing of 10 km have been analyzed for this manuscript, nextGEMS also performed simulations with a horizontal grid spacing of 2.8 and 5 km integrated over shorter time periods. The lessons learned

in nextGEMS are transferred to other projects, which conduct climate simulations with horizontal grid spacings up to 1.25 km. In that sense, nextGEMS is only the first step in a new era of km-scale climate simulations.

This paper is structured as follows. In Section 2, we provide an overview of the project concept and models, ICON and IFS-FESOM. In Section 3, we report the main changes made in both models over the four cycles. In Section 4, we discuss the four questions stated above and examine the realism of our km-scale simulations. In Sections 5 and 6, we summarize the development of the past three years and provide an outlook for the future of nextGEMS.

## 2 nextGEMS concept and models

### 2.1 Concept

nextGEMS followed an innovative approach to create an integrated community of experts from different domains, including climate science, model development, and high-performance computing (HPC). The model development was structured into four cycles from October 2021 to March 2024. The cycles had an average length of about 8 months and facilitated the fast-paced evolution of the two models, ICON and IFS-FESOM. Each cycle was marked by a coordinated set of simulations with both models and by a hackathon, where more than 100 project participants came together for a week of interactive coding, debugging, plotting, and technical and scientific discussions. The hackathons turned the model development into a collaborative endeavor, with close interaction between the modeling centers and their partner institutions in 14 countries. The hackathons were also used as opportunities to engage with industry stakeholders and show the potential of such km-scale climate simulations for applications such as wind and solar energy or agriculture and fisheries. The interaction patterns between participants were analysed across hackathons, revealing an increase in inter-institutional cooperation and cross-disciplinary exchange over the three years. The outcome and feedback from the hackathons were also critical in the preparation of the next cycle, always aiming to produce more scientifically sound and computationally efficient simulations over longer periods. Figure 1 shows the timeline of the whole project from Cycle 1 to 4. The simulation times evolved from a few weeks with throughputs of about 10 SDPD to three decades with throughputs of about 500 SDPD.

The simulations were produced and stored on the HPC-system Levante (HLRE-4 Levante, 2024). The participants of hackathons worked directly on Levante instead of copying large amounts of simulation data to individual systems. We took several steps to make the data analysis user-friendly and model-agnostic. Most simulation data was provided via Intake catalogs, a Python package for searching and loading data (https://github.com/intake/intake). From Cycle 2 onwards, the simulations were published on the World Data Center for Climate (WDCC). And from Cycle 3 onwards, the simulations were unified into a single Intake catalog (https://data.nextgems-h2020.eu/catalog.yaml). The output of ICON was stored as Zarr datasets (https://github.com/zarr-developers/zarr-python), and the output of IFS-FESOM was indexed using gribscan (Kölling et al., 2024), a tool to scan gridded binary (GRIB) files and create Zarr-compatible indices such that users can access the underlying GRIB files as Zarr datasets. In addition, example notebooks for an initial analysis were provided via the easygems website (https://easy.gems.dkrz.de and https://github.com/nextGEMS).

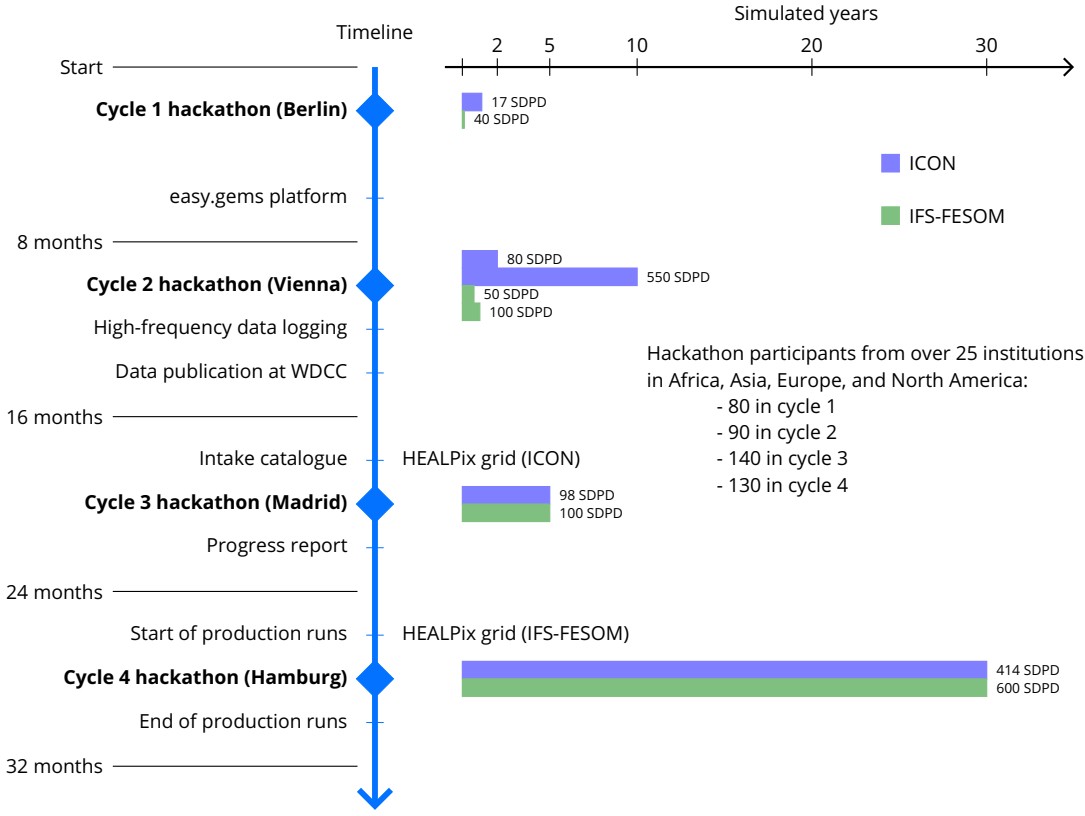

**Figure 1.** Timeline of nextGEMS from Cycle 1 to 4. The vertical axis shows the development cycles, including hackathons and milestones, and the horizontal axis shows the simulated years, including throughputs in simulated days per day (SDPD).

The two models developed in nextGEMS, ICON and IFS-FESOM, can simulate the atmosphere, ocean and sea ice, and land and their interactions at km-scales. Moreover, they can include additional components of the Earth system, such as the carbon cycle (and aerosols). For this reason, we refer to ICON and IFS-FESOM as km-scale Earth system models, even

though the simulations presented here were conducted without any carbon cycle module. Figure 2 shows an overview of how these components and their interactions are represented in each model. The two models are complementary. While both models simulate the Earth system, ICON can be characterized as a research model with a minimal set of parameterization schemes, whereas IFS-FESOM can be characterized as an operational model with elaborate parameterization schemes and tuning methods. The key features of both models are described in the next two sections and the developments over the four

cycles are summarized in Section 3 (and Appendix A).

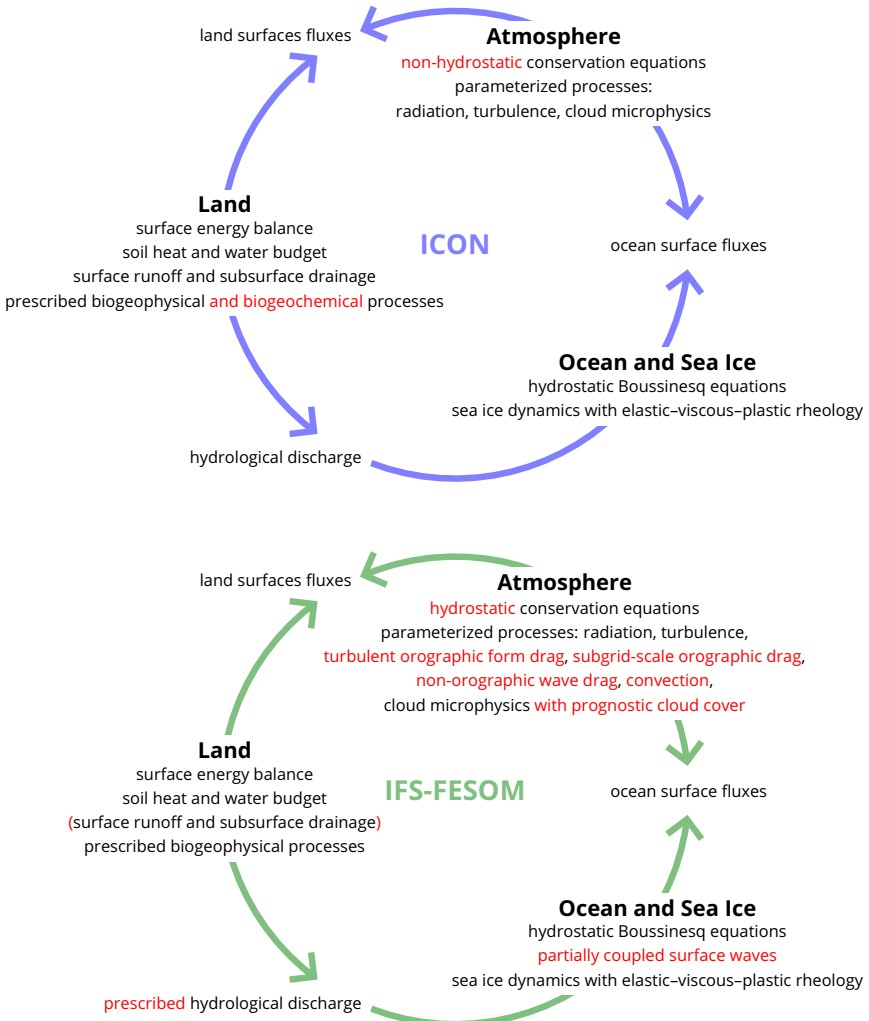

**Figure 2.** Overview of the Earth system models ICON and IFS-FESOM. Differences between the two models are highlighted in red.

## 2.2 Models

### 2.2.1 ICON

The ICOsahedral Non-hydrostatic model (ICON) is developed by the ICON Partnership. A detailed first description of the km-scale version was presented by Hohenegger et al. (2023). In ICON, the Earth system is divided into three coupled components:

atmosphere, land, and ocean. All three components are discretized with an icosahedral-triangular C grid (Zängl et al., 2015).

The atmosphere is modeled with non-hydrostatic equations (Zängl et al., 2015). It includes tendencies from parameterized processes: turbulence (Dipankar et al., 2015; Lee et al., 2022), radiation (Pincus et al., 2019), and cloud microphysics (Baldauf et al., 2011). It is discretized with terrain-following levels in the vertical (Leuenberger et al., 2010). In addition, it includes

the one-moment aerosol module HAM-lite developed in nextGEMS (Weiss et al., 2025). HAM-lite represents aerosols as an ensemble of log-normal modes with prescribed sizes and compositions to reduce the computational costs. The aerosol modes are transported as prognostic tracers and are coupled with the parameterized processes mentioned above. The land is represented with the land surface model Jena Scheme for Biosphere-Atmopshere Coupling in Hamburg (JSBACH; Reick et al., 2021) and interacts with the atmosphere via surface fluxes and with the ocean via hydrological discharge. It is discretized with multiple soil layers (Reick et al., 2021). The ocean is modeled with hydrostatic Boussinesq equations (Korn et al., 2022) and interacts with the atmosphere via the Yet Another Coupler (YAC; Hanke et al., 2016). It is discretized with variable levels in the vertical that follow the movement of the ocean surface (Korn et al., 2022). The ocean dynamics are spun up as described in Hohenegger et al. (2023) for Cycle 1 and as described in Appendix A for later cycles. Lastly, the sea ice dynamics are modeled based on elastic-viscous-plastic rheology (Danilov et al., 2015).

### 2.2.2   IFS-FESOM

The Integrated Forecasting System (IFS) is developed and maintained by the European Centre for Medium-Range Weather Forecasts (ECMWF) in collaboration with institutions in its member and cooperating states. A detailed description of the model can be found in the official documentation (ECMWF, 2023a, b). The IFS is centered around the atmosphere coupled with other components of the Earth system such as land, ocean, and sea ice. The atmosphere and land are discretized with an octahedral-reduced Gaussian (TCo) grid (Malardel and Wedi, 2016). The ocean is discretized for nextGEMS with either a triangulated 0.25-degree grid (tORCA025) or an eddy-resolving (NG5) grid (Rackow et al., 2025b).

The atmosphere is modeled with hydrostatic equations and includes parameterization schemes for radiation, turbulence, turbulent orographic form drag, subgrid-scale orographic drag, non-orographic wave drag, convection, and cloud microphysics including a prognostic cloud cover (ECMWF, 2023b). It is discretized with hybrid levels in the vertical, transitioning from terrain-following coordinates at lower levels to pressure-level coordinates at higher levels (Simmons and Strüfing, 1983). The land is represented with the land surface model ECLand (Boussetta et al., 2021) and discretized with four soil layers. In its current configuration, ECLand computes local runoff, but it does not route the runoff nor provide it as freshwater input into the ocean. Instead, the hydrological discharge is prescribed for all coastal points. In Cycle 1, the ocean is represented with the FESOM2.1 model (Wang et al., 2014; Scholz et al., 2019), developed and maintained by the Alfred Wegener Institute, Helmholtz Centre for Polar and Marine Research (AWI). It is discretized with the tORCA025 grid using a parameterization for mesoscale eddies (Gent and Mcwilliams, 1990; Gent et al., 1995). In the later cycles, the ocean is updated to FESOM2.5 and is discretized with the NG5 grid using an eddy-resolving resolution of 5 kilometers (Rackow et al., 2025b). In all cycles, the vertical dimension is discretized with Arbitrary-Lagrangian-Eulerian coordinates. The ocean dynamics are spun up in stand-alone mode for 5 years with atmospheric forcing from the ERA5 reanalysis (Hersbach et al., 2020) before the coupled IFS-FESOM simulations are started. Lastly, waves are represented with the ecWAM model (Janssen, 2004), and sea ice is represented with the FESIM model (Danilov et al., 2015, 2017).

### 2.2.3 Output convergence

A large effort was made to harmonize the model output and to facilitate an easy comparison between the two models. As an important step towards this end, ICON and IFS-FESOM both provided their output on a HEALPix grid in Cycle 4 (Górski et al., 2005). HEALPix stands for Hierarchical Equal Area isoLatitude Pixelation of a sphere. The pixels of such a grid all have the same area and are located on lines of constant latitude. In addition, the hierarchical tessellation of such a grid makes it possible to visualize and process the data on different resolutions (https://healpix.sourceforge.io). In ICON, the HEALPix grid was incorporated with an improved version of the YAC coupler (Hanke et al., 2016) and the newly developed HIOPY module (https://gitlab.gwdg.de/ican/hiopy/). In IFS-FESOM, the HEALPix grid was incorporated with the multIO framework (Sarmany et al., 2024) and the MIR package (Maciel et al., 2017). This achievement was made possible thanks to synergistic work with the Climate Adaptation Digital Twin of Destination Earth. Apart from the HEALPix grid, developers of ICON and IFS-FESOM worked closely together to provide a minimum set of common variables, to output atmospheric variables at a common set of pressure levels, and to output variables on a common, height-dependent frequency. From Cycle 2 onwards, IFS-FESOM provided some post-processed output of general interest, such as monthly means and coarser output on a regular 0.25-degree grid (Rackow et al., 2025b). From Cycle 3 onwards, the output was computed on the fly by multIO (Sarmany et al., 2024) and MIR (Maciel et al., 2017) before writing it to disk.

## 3 Simulations

In this section, we describe the important characteristics of each cycle, focusing on the simulation period, computational throughput, and model biases in terms of mass and energy conservation. To refer to the simulations, we use the following conventions: ICON-C$X$ and IFS_F-C$X$, in which $X$ indicates the cycle and IFS_F stands for IFS-FESOM. In the case of two simulations from the same cycle, we added the letters A and B, e.g., ICON-C2-A and ICON-C2-B or IFS_F-C2-A and IFS_F-C2-B. Table 1 provides an overview of the simulations from Cycles 1 to 4. All simulations started on 20 January 2020. The last column shows whether the simulation was energetically consistent, as defined in Section 1. The simulation data was published on the World Data Center for Climate (Wieners et al., 2023; Koldunov et al., 2023; Wieners et al., 2024).

While there is a clear increase in throughput over all cycles, the input/output (I/O) operations in IFS_F-C2-A and IFS_F-C2-B were extremely expensive as there were no dedicated I/O servers. The computational throughput was, therefore, less than expected for 864 nodes. This has been solved with the introduction of MultIO (Sarmany et al., 2024) in IFS and FESOM in Cycle 3 and Cycle 4, as explained in Section 3.3 and Section 3.4, respectively.

### 3.1 Cycle one

The first cycle started with the ICON and IFS-FESOM versions presented by Hohenegger et al. (2023) and Rackow et al. (2025b), respectively. The simulation with ICON, referred to as ICON-C1, used a horizontal grid of 5 km in the atmosphere and ocean. In addition, there were 90 vertical levels in the atmosphere, 128 vertical levels in the ocean, and 5 soil layers in the

**Table 1.** Overview of all simulations from nextGEMS Cycle 1 to 4 performed with ICON and IFS-FESOM (IFS_F). $\Delta x_A$ and $\Delta x_O$ indicate the horizontal resolution of the atmosphere and ocean, respectively. Period indicates the integration time. Nodes refers to the number of CPU nodes on Levante (128 cores in total, 256 GB main memory) or an equivalent number of nodes (indicated with $^*$). Throughput refers to the simulated days per day (SDPD), normalized by the number of nodes (SDPD node$^{-1}$). The last two columns show the atmospheric radiative forcing and whether the simulation is energetically consistent. The simulation data of Cycles 2 to 4 was published on the World Data Center for Climate (Wieners et al., 2023; Koldunov et al., 2023; Wieners et al., 2024).

| | $\Delta x_A$ (km) | $\Delta x_O$ (km) | Period | Nodes | Throughput (SDPD, SDPD node$^{-1}$) | Forcing | Energetically consistent |
|---|---|---|---|---|---|---|---|
| Cycle 1 | | | | | | | |
| ICON-C1 | 5 | 5 | 406 days | 100* | 17, 0.17 | 2020 | No |
| IFS_F-C1 | 4.4 | 25 | 40 days | 20* | 40, 2 | 2020 | No |
| Cycle 2 | | | | | | | |
| ICON-C2-A | 5 | 5 | 2 years | 400 | 80, 0.2 | 2020 | No |
| ICON-C2-B | 10 | 10 | 10 years | 400 | 550, 1.375 | 2020 | No |
| IFS_F-C2-A | 2.8 | 5 | 8 months | 864* | 50, 0.057 | 2020 | No |
| IFS_F-C2-B | 4.4 | 5 | 1 year | 864* | 100, 0.116 | 2020 | No |
| Cycle 3 | | | | | | | |
| ICON-C3 | 5 | 5 | 5 years | 530 | 98, 0.185 | 2020 | No |
| IFS_F-C3 | 4.4 | 5 | 5 years | 269 | 100, 0.372 | 2020 | Yes |
| Cycle 4 | | | | | | | |
| ICON-C4 | 10 | 5 | 30 years | 464 | 414, 0.892 | SSP3-7.0 | Yes |
| IFS_F-C4 | 9 | 5 | 30 years | 269 | 600, 2.23 | SSP3-7.0 | Yes |

land. The time step was 40 s for the atmosphere (and land) and 80 s for the ocean. Note that the same simulation was presented by Hohenegger et al. (2023) under the name G_AO_5km. ICON-C1 was integrated over one year on the Mistral supercomputer of DKRZ. The simulation with IFS-FESOM, referred to as IFS_F-C1, was based on IFS CY46R1 (ECMWF, 2019) and it used

a horizontal grid spacing of 4.4 km in the atmosphere and 25 km in the ocean (a triangulated version of NEMO's ORCA025 grid for use with FESOM, coined 'tORCA025'). IFS_F-C1 was integrated over 40 days on the Cray supercomputer of ECMWF. Both Mistral and Cray supercomputers were decommissioned in 2021 and replaced with Levante (HLRE-4 Levante, 2024) and Atos (ECMWF, 2024), respectively.

ICON-C1 and IFS_F-C1 both showed large imbalances in the energy budget. In ICON-C1, the imbalance in the atmospheric
energy budget of about $-4\,\mathrm{W\,m^{-2}}$ caused a drift in the global mean surface temperature of about $-2\,\mathrm{K}$ (Mauritsen et al., 2022; Hohenegger et al., 2023). In IFS_F-C1, the imbalance in the energy budget of $6.4\,\mathrm{W\,m^{-2}}$ (and smaller at coarser resolutions) was related to a leak in the water budget in the atmosphere. This leak caused a spurious increase in the total precipitation of about $4.6\,\%$ (Rackow et al., 2025b). Bugs related to the energy and water imbalance in ICON and IFS were already present in previous simulations at coarser horizontal resolutions, and another configuration in the case of ICON, without causing

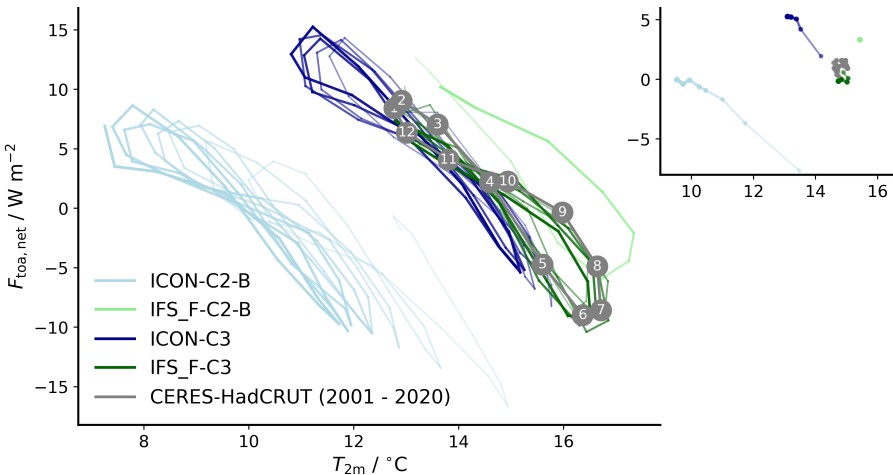

**Figure 3.** Annual cycles of the monthly mean 2-meter temperature and top-of-atmosphere radiation balance from ICON-C2-B, ICON-C3, IFS_F-C2-B, and IFS_F-C3. Note that the radiation balance is computed as the difference of the downwelling and upwelling radiation, i.e., $F_{\text{toa,net}} = F_{\text{toa,sw}}^{\downarrow} - F_{\text{toa,sw}}^{\uparrow} - F_{\text{toa,lw}}^{\uparrow}$ (Mauritsen et al., 2022). The widths and opacities of the lines increase with time. Gray line shows the observational reference averaged over 2001 to 2020, i.e., 2-meter temperature from HadCRUT (Morice et al., 2021) and radiation balance from CERES-EBAF (Loeb et al., 2018). The months are labeled with numbers. The corresponding annual means are shown in the top-right inset. Like for the annual cycles, the widths and opacities of the lines increase with time.

any evident problems. In other words, the impacts of such bugs were negligible at coarse spatiotemporal scales and only became evident at the much finer spatiotemporal scales simulated in nextGEMS – an important lesson learned. A more detailed discussion on the identification and resolution of bugs in nextGEMS was presented by Proske et al. (2024). In the case of IFS, the proposed solutions were first tested at coarser resolutions before being implemented in its km-scale version. The coarsest resolution for scientific tests was 28 km. In the case of ICON, the proposed solutions were directly tested at finer resolutions of 10 km. Only particular technical problems were addressed with tests at coarser resolutions of 140 km. In addition, simplified idealized cases were used to isolate and debug those parts of the code that caused the energy and water leak.

On the technical side, the throughput in ICON-C1 was 17 SDPD on 400 nodes of the Mistral supercomputer, roughly corresponding to 100 nodes of Levante, whereas the throughput in IFS_F-C1 was 40 SDPD using 60 nodes of the Cray supercomputer, roughly corresponding to 20 nodes of Levante. With these numbers, ICON could simulate 0.17 SDPD per node, 10-fold less than IFS (2 SDPD per node). As a rule of thumb, a possible explanation is the coarse horizontal grid spacing in the ocean module in IFS. The different machines used to simulate ICON-C1 and IFS_F-C1 could also be an additional reason for higher SDPD per node in IFS. Thus, aside from the large energy imbalance, the small computational throughputs were also the main obstacles to the upcoming decadal simulations.

## 3.2 Cycle two

In the second cycle of IFS-FESOM, the IFS version was upgraded to CY47R3 (ECMWF, 2021). This included, among other changes, the inclusion of a 5-layer snow scheme, compared to the previously used bulk scheme. In addition, the atmospheric imbalance of water and, therefore, energy was significantly reduced to less than $1\,\mathrm{W\,m^{-2}}$. This was done by activating global tracer mass fixers (Rackow et al., 2025b). Such modifications also improved the NWP configuration of IFS-NEMO at ECMWF (Rackow et al., 2025b) and were, therefore, incorporated into its operational version CY48r1. Two IFS-FESOM simulations,

IFS_F-C2-A and IFS_F-C2-B, were provided. Both simulations shared a common horizontal grid spacing in the ocean ($\Delta x = 5\,\mathrm{km}$) but used different grid spacings in the atmosphere. IFS_F-C2-A used a horizontal grid spacing of 2.8 km and was integrated for 8 months, whereas IFS_F-C2-B used a horizontal grid spacing of 4.4 km and was integrated for 1 year. Both simulations were subject to a warming trend on the order of $1\,\mathrm{K\,y^{-1}}$ as shown in Figure 3.

In the second cycle of ICON, the energy leak in Cycle 1 was traced back to bugs in the dynamical core, cloud microphysics,

and surface fluxes within the turbulence scheme. Bugs in the microphysics and turbulence were resolved, and the energy imbalance was reduced. Besides these fixes, the new radiation scheme RTE-RRTMGP (Pincus et al., 2019) was implemented, and a cloud inhomogeneity factor was introduced to tune the radiation balance at the top of the atmosphere, as discussed further in Section 4.1. In the ocean module, a new vertical coordinate system with thinner surface levels was used. Further details on the ICON development in this cycle are summarized in Appendix A. After 8 test runs, two simulations were provided. ICON-

C2-A used a horizontal grid spacing of 5 km and was integrated for 2 years, whereas ICON-C2-B used a horizontal grid spacing of 10 km and was integrated for 10 years. Both simulations were subject to a cold drift, as shown in Figure 3.

IFS_F-C2-A and IFS_F-C2-B were both performed on the Atos supercomputer of ECMWF, where one node is roughly equivalent to one node in Levante. The throughput in IFS_F-C2-A (2.8 km) was 50 SDPD on 864 nodes, giving a normalized throughput of 0.057 SDPD per node. Coarsening the resolution by roughly a factor of 2 in IFS_F-C2-B (4.4 km) increased the

240 throughput by a factor of 2 on the same number of nodes, resulting in a normalized throughput of 0.116 SDPD per node. The 17-fold reduction in the normalized throughput in IFS_F-C2-B compared to IFS_F-C1 is explained by several factors. In Cycle 2, IFS-FESOM introduced an eddy-resolving ocean. This means an 8-fold increase in grid points and a 5-fold smaller timestep, resulting in 40-fold higher costs for the ocean. More output variables were written in Cycle 2 with a higher spatial and temporal resolution. Moreover, the IO configuration for IFS-FESOM was suboptimal as it did not make use of any IO servers yet. In

addition, full ocean support for hybrid MPI-OpenMP parallelization was only supported with the release of FESOM version 2.5 (used from Cycle 3 onwards). We also decided to make use of a large number of available nodes on the Atos supercomputer at the time, despite the lack of an optimized IO configuration. Thus, even with a considerably greater number of resources in Cycle 2 compared to Cycle 1, which sped up the computing part of the model, the total throughput of IFS-FESOM, including IO, did not scale accordingly, and a large number of used nodes, therefore, reduces the normalized throughput considerably.

ICON-C2-A and ICON-C2-B were both performed on the Levante supercomputer. The throughput in ICON-C2-A (5 km) was 80 SDPD on 400 nodes, meaning 0.2 SDPD per node. Coarsening the resolution by a factor of 2 in ICON-C2-B (10 km) increased the throughput by a factor of 7 on the same number of nodes, increasing the normalized throughout to 1.375 SDPD

per node. With the same resolution in ICON, simulations in Cycle 1 and 2 have the same throughput per node, meaning that the increase in throughput was due to the use of more nodes.

## 3.3 Cycle three

In the third cycle of IFS-FESOM, the base version of IFS was upgraded to CY48R1 (ECMWF, 2023a, b; Rackow et al., 2025b). On top of that, the representation of uncertain cloud and microphysical processes was optimised to yield improved shortwave and longwave TOA fluxes and, as a result, a top-of-atmosphere radiation budget within observational uncertainty. The reduced cloud base mass-flux approach was introduced to increase the convective organization in simulations at 4.4 km resolution. In the land module, land use/land cover information was revised to describe km-scales better (Boussetta et al., 2021), and a parametrization for urban processes was added (McNorton et al., 2021, 2023) in line with developments for CY49R1. Further river hydrology was enabled for selected simulations by 1-way coupling precipitation and runoff output of IFS-FESOM to the Catchment-based Macro-scale Floodplain (CaMa-Flood) model v4.1 (Yamazaki et al., 2014, 2011), generating river discharge and flooded fraction. FESOM was upgraded to version 2.5 with significant changes in the atmosphere-ocean coupling, such as the coupling of ocean surface velocities and accounting for the enthalpy of snow falling into the ocean (Rackow et al., 2025b). With all these modifications, IFS-FESOM was integrated in time for 5 years using a horizontal grid spacing of 4.4 km in the atmosphere and 5 km on average in the ocean (IFS_F-C3). In contrast to previous cycles, IFS_F-C3 was not subject to drift in the surface temperature compared to observations, indicating an energetically consistent climate (Figure 3). Moreover, the radiation fluxes at the top of the atmosphere were also close to observations (Rackow et al., 2025b).

In the third cycle of ICON, the energy imbalance was further reduced by fixing bugs in the cloud microphysics and turbulence scheme. In the calculation of surface fluxes, the heat capacity at constant pressure was replaced by the heat capacity at constant volume. ICON computes those fluxes under the assumption of constant volume (not pressure). This change reduced the amount of energy transferred from the surface to the atmosphere by about $29\%$. Like in the previous cycle, the cloud inhomogeneity factor was used to tune the radiation balance. Further changes in the ocean and land modules are summarized in Appendix A. After 27 test runs, ICON was integrated for 5 years using a horizontal grid spacing of 5 km in the ocean and atmosphere. ICON-C3 exhibited a cooling trend of about $0.33 \, \mathrm{K \, y^{-1}}$, as shown in Figure 3. This cold drift could be traced back to remaining energy leaks in the dynamical core.

All simulations with IFS-FESOM were performed on the Levante supercomputer from Cycle 3 onwards. The throughput per node showed a 3-fold increase in Cycle 3 (0.372 SDPD per node) compared to Cycle 2 (0.116 SDPD per node), explaining the similar throughput (100 SDPD) even if IFS_F-C3 used a third of the resources of IFS_F-C2-B. The resource efficiency was increased substantially by optimizing the I/O operations, and from Cycle 3 onwards, I/O servers via multIO were introduced in IFS. Moreover, less 3D ocean output was written in Cycle 3 compared to Cycle 2 (only the upper 300 m at 3-hourly frequency), which dramatically reduced the time spent for I/O in IFS-FESOM. In ICON, the throughput per node in Cycle 3 (0.185 SDPD per node) was roughly similar to Cycle 2 (0.2 SDPD per node). Thus, the use of slightly more resources in Cycle 3 (530 nodes) compared to Cycle 2 (400 nodes) increased the throughput from 80 to 98 SDPD.

## 3.4 Cycle four

In the fourth cycle of IFS-FESOM, given the success of providing energetically consistent simulations in Cycle 3, the last changes needed were mostly related to preparing input for climate projections. In addition, only daily output was written for ocean fields, in line with IFS-FESOM simulations for the Climate Change Adaptation Digital Twin in Destination Earth.

IFS_F-C4 used a horizontal grid spacing of 9 km in the atmosphere and 5 km in the ocean and was integrated for 30 years. Additionally, CaMa-Flood was run 1-way coupled to generate river hydrology on the respective period.

In Cycle 4, the big challenge in ICON was to simulate an energetically consistent climate. To this end, the following changes in the atmosphere module were made. The diffusivity in the momentum equation of the dynamical core was modified to account for the persistent but small leak in internal energy in the dynamical core. The old turbulent mixing scheme VDIFF

was upgraded to enable both explicit and implicit numerical solvers, to replace the implicit atmosphere-surface coupling with an explicit coupling, and to shift the diffused thermal quantity from dry static energy to internal energy. This new upgrade is called TMX, and it contributed to a further reduction of the energy leak at the surface. Lastly, the cloud inhomogeneity factor was linked with the lower tropospheric stability to better account for the cloud type, as discussed in Section 4.1. Further details on the development of ICON-C4 are summarized in Appendix A. Furthermore, the implementations to reduce the energy leak

in ICON are now taken as a base for future simulations. After numerous short test runs, ICON-C4 was integrated for 30 years using a horizontal grid spacing of 10 km in the atmosphere and 5 km in the ocean.

ICON-C4 and IFS_F-C4 both used greenhouse gas concentrations, including ozone, following the Coupled Model Intercomparison Project Phase 6 (CMIP6) for the forcing time series of a SSP3-7.0 scenario (O'Neill et al., 2016). In addition, ICON-C4 used the time-varying Max Planck Institute Aerosol Climatology (MACv2-SP; Stevens et al., 2017) for anthropogenic aerosols

together with the Stenchikov climatology from 1999 for volcanic aerosols (Stenchikov et al., 1998). IFS_F-C4 used the time-varying tropospheric climatology provided by the CONsistent representation of temporal variations of boundary Forcings in reanalysES and Seasonal forecasts (CONFESS) project (Stockdale et al., 2024), in which volcanic aerosols from 1850 with no large eruptions were applied to all years after 2014.

In view of the project timeline and less than desired access to computational resources, a compromise had to be made between

the model resolution and simulation length for the production simulations. This led to 30-year simulations at resolutions of about 10 km in the atmosphere and 5 km in the ocean. In ICON-C4, the 10 km grid spacing of the atmosphere allowed us to reach a throughput per node of 0.892 SDPD per node, and using 464 nodes (18% of Levante's nodes), the total throughput was 414 SDPD. In IFS_F-C4, the 9 km grid spacing in the atmosphere, together with further improvements in I/O operations by using multIO for both IFS and FESOM, gave a throughput per node of 2.23 SDPD per node. Using 269 nodes (9% of Levante's

nodes), IFS_F-C4 simulated 600 SDPD.

Thus, in less than three years of model development, nextGEMS developed two models with an energetically consistent climate, which can be a persistent issue in established climate models (e.g. Sanderson and Rugenstein, 2022), and with a competitive throughput. Taken together, multidecadal and global km-scale climate simulations, in which land, atmosphere,

and ocean are coupled, have now become a reality. This opens new ways to analyze regional atmospheric, oceanic, and land
processes on global scales and their changes in a global warming context.

## 4 Insights into the realism of km-scale simulations

In this part, we examine the realism of the Cycle 4 simulations outlined in Section 3.4. To create a storyline, we discuss four central questions on the radiation balance, key features of mean climate, local- to synoptic-scale phenomena, and time scales of regional patterns, as introduced in Section 1. The reader should note that robust and definite answers to these questions will form gradually as separate studies from the different scientific groups of nextGEMS emerge.

### 4.1 Radiation balance

As outlined in Section 3, it was possible to improve the conservation of mass and energy and eliminate the large drifts in the near-surface temperature observed in the early simulations. This does not imply necessarily, however, that the energy balance at the top-of-atmosphere agrees with observations. In both ICON and IFS-FESOM, cloud radiative properties and formation mechanisms were adjusted to reduce the remaining biases. However, the tuning strategies in ICON and IFS-FESOM differed. The low cloud cover, for example, was too high in ICON but too low in IFS-FESOM.

In ICON, the cloud cover was adjusted in two steps targeting the turbulent mixing and cloud brightness. First, we adjusted the mixing in the Smagorinsky scheme, which depends on the Richardson number, following the formulation of Louis (1979). This adjustment allows some mixing or entrainment in situations where the traditional Smagorinsky scheme would yield no mixing. Mixing at large Richardson numbers controls stratiform boundary layer clouds but has only small effects on trade wind clouds. Second, we accounted for the lack of a cloud fraction scheme through a cloud inhomogeneity factor ($\zeta$), which depends on the lower-tropospheric stability (LTS). LTS is defined as the difference in potential temperature between the free troposphere and the surface and is strongly correlated with the stratiform low cloud cover (Wood and Bretherton, 2006). In theory, the inhomogeneity factor is equal to one for fully-resolved, homogeneous clouds and is less than one for partially-resolved, inhomogeneous clouds, as discussed by (Cahalan et al., 1994; Mauritsen et al., 2012). It scales down the cloud water and ice before the shortwave fluxes are calculated in the radiation scheme. In ICON, the inhomogeneity factor acts only on the liquid clouds, whereas the ice clouds remain unchanged. It increases non-linearly from about $0.4$ ($\zeta_{\min}$) at a lower-tropospheric stability of $0\,\mathrm{K}$ to about $0.8$ ($\zeta_{\max}$) at a lower-tropospheric stability of $30\,\mathrm{K}$, i.e.,

$$\zeta = \zeta_{\min} + (\zeta_{\max} - \zeta_{\min})\left(1 - \frac{\arctan 2\,(c_1, \mathrm{LTS} - c_2)}{\pi}\right), \tag{1}$$

where $\zeta_{\min} = 0.4$, $\zeta_{\max} = 0.8$, $c_1 = 2\,\mathrm{K}$ and $c_2 = 20\,\mathrm{K}$ are tuning parameters.

In IFS-FESOM, the cloud cover is a prognostic variable of the cloud microphysics scheme (ECMWF, 2023b). It shows a resolution and time step dependence. Overall, it is smaller at higher resolution with shorter time steps than at coarser resolutions with longer time steps. To find a configuration that is close to observations of both longwave and shortwave radiative fluxes from CERES-EBAF (Loeb et al., 2018), the cloud cover was modified as documented in Rackow et al. (2025b) by

    - reducing the inhomogeneity enhancement factor for accretion from 3 to 2,

   - reducing the cloud edge erosion from $6 \cdot 10^{-6}$ to $4 \cdot 10^{-6}$, and

   - assuming a constant effective cloud spacing, following recommendations by Fielding et al. (2020).

Overall, these changes led to an increase in the low cloud cover. In addition, the high cloud cover was increased in areas with strong deep convective activity by

    - decreasing a threshold that limits the minimum size of the ice effective radius from 60 to 40, in agreement with observational evidence

   - and changing from cubic to linear departure point interpolation in the semi-Lagrangian advection scheme for all water species except vapour.

In combination, these changes in the cloud cover in IFS-FESOM increased the outgoing shortwave radiation by about $5\,\mathrm{W\,m^{-2}}$, while decreasing outgoing longwave radiation by about $3\,\mathrm{W\,m^{-2}}$. This led to a well-balanced radiation budget at the TOA with both shortwave and longwave fluxes in realistic ranges.

Figure 4 shows the time series of the monthly mean 2-meter temperature (Figure 4a) and annual cycles of the monthly mean 2-meter temperature and top-of-atmosphere radiation balance (Figures 4b,c) over the whole simulation period. The annual means of the near-surface temperature and radiation balance are shown in Figures 4b,c. The corresponding observations from CERES (Loeb et al., 2018) and HadCRUT (Morice et al., 2021) are shown as well. As discussed by Mauritsen et al. (2012, 2022), the annual cycles are shaped like a figure eight. In the first half of the year, the radiation balance decreases while the near-surface temperature increases. In the second half, the radiation balance increases while the near-surface temperature decreases. The near-surface temperature lags behind the radiation balance, as would be expected from the large heat capacity of the Earth system. Overall, the values of IFS-FESOM agree better with observations than the values of ICON. The near-surface temperatures are $13.83\,^{\circ}\mathrm{C}$ for ICON, $14.94\,^{\circ}\mathrm{C}$ for IFS-FESOM, and $14.91\,^{\circ}\mathrm{C}$ for HadCRUT, averaged over the first four years from 2020 to 2023. And the top-of-atmosphere radiation balances are $0.25\,\mathrm{W\,m^{-2}}$ for ICON, $0.51\,\mathrm{W\,m^{-2}}$ for IFS-FESOM, and $1.46\,\mathrm{W\,m^{-2}}$ for CERES. Moreover, the near-surface temperature of ICON cools in contrast to IFS-FESOM. We assume that this initial cooling is related to the adjustment of the atmosphere to the ocean, which is spun up with ERA5 forcing. In the subsequent years, the radiation balance is positive, and the near-surface temperature increases in line with the SSP3-7.0 scenario in both ICON and IFS-FESOM. A detailed analysis of the shortwave reflectivity of stratocumulus clouds, which are only partially resolved in km-scale Earth System models, is presented in Section 4.3.2.

## 4.2 Key features of mean climate

Our next question is whether a simulation with an energetically consistent climate, together with solving the oceanic and atmospheric flows on the km-scale, can have an adequate representation of the Earth's mean climatic features. To address this, we select two key large-scale features in the energetics of the climate system: (i) the tropical rainbelt, and (ii) the pattern of

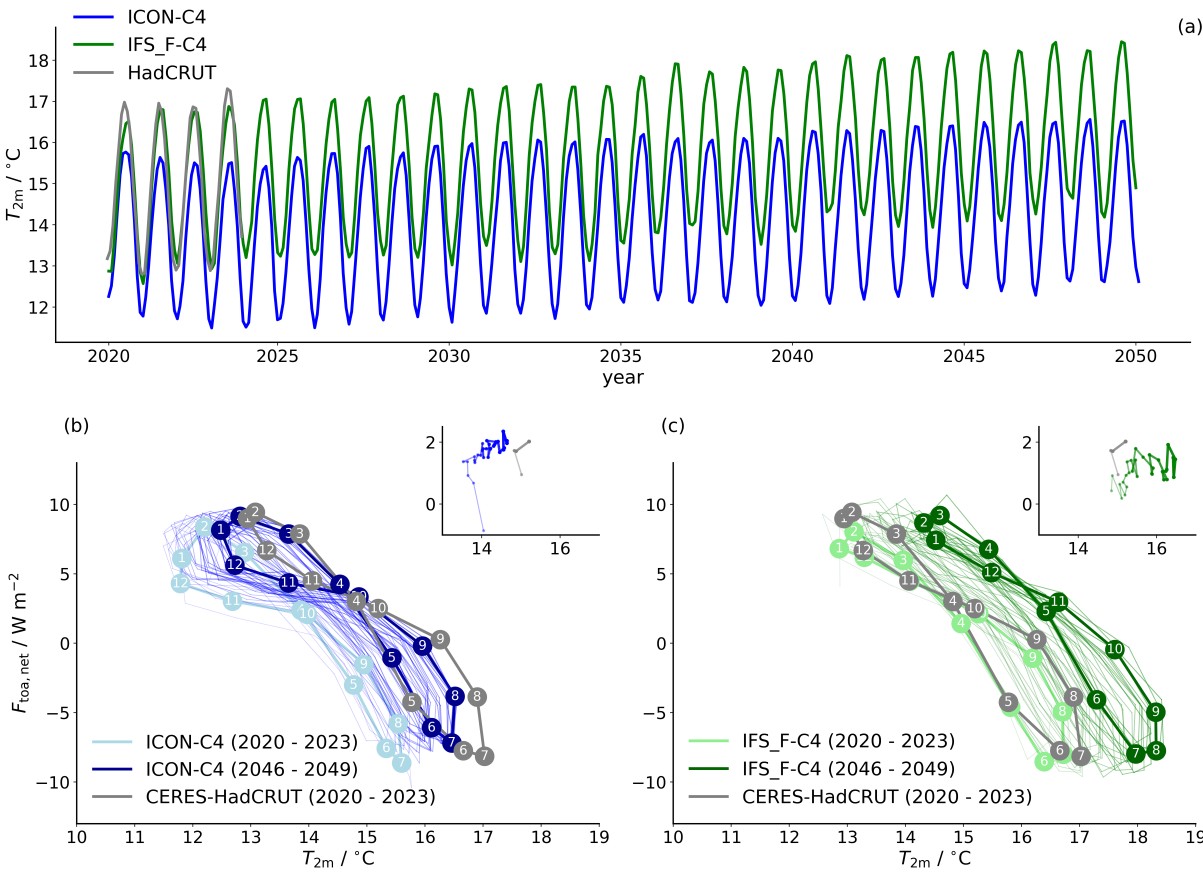

**Figure 4.** Time series of the monthly mean 2-meter temperature from ICON-C4 and IFS_F-C4 (a). Annual cycles of the monthly mean 2-meter temperature and top-of-atmosphere radiation balance from ICON-C4 (b) and IFS_F-C4 (c). Thick lines indicate averages over 2020 to 2023 (light colors) and 2046 to 2049 (dark colors), where months are labeled with numbers. Thin lines indicate individual cycles from 2020 to 2049, where widths and opacities increase with time. Gray lines show the observational reference, i.e., 2-meter temperature from HadCRUT (Morice et al., 2021) and radiation balance from CERES-EBAF (Loeb et al., 2018). The corresponding annual means are shown in the top-right insets. As for the annual cycles, the widths and opacities increase with time.

sea surface temperature (SST). The role played by km-scale processes in convection gives an additional reason to analyze the representation of the tropical rainbelt, which has been notoriously difficult to reproduce in coarse-resolution Earth system models (Mechoso et al., 1995; Lin, 2007). The analysis of the SST patterns, which coarse-resolution Earth system models fall short in reproducing, is important because being able to capture it has emerged as an important factor in determining climate sensitivity. The tropical rainbelt is here defined as the quantile 80 of the yearly precipitation mean in the tropics, i.e., it corresponds to the 20% wettest region in the tropics (30°S-30°N; see Segura et al., 2022). Using ICON-C4 and IFS_F-C4, respectively, the yearly precipitation mean is computed from the first five complete years of integration, 2021-2025. This pragmatic choice is made here to allow nearly one year of spinup, from 20th January to 31st December 2020, and because

computing the yearly mean for the entire 30-year period would include the impact of the SSP3-7.0 forcings on the climate.
In these five years, the El Niño index 3.4 is in near-neutral conditions in both simulations, with the exception of one year in IFS_F-C4. This means that the biases and the correct representation of large-scale features are not affected by the ENSO variability.

The structure of the terrestrial tropical rainbelt is encouragingly similar between both simulations and observations from the precipitation satellite product Integrated Multi-satellitE Retrievals for GPM (IMERG; Huffman et al., 2019) (Figure 5). The location is well reproduced, albeit with a reduced area in both simulations over South America. Regarding the tropical ocean, ICON-C4 and IFS_F-C4 show a similar structure of the tropical rainbelt in the Eastern Pacific that matches IMERG. However, a less consistent picture between both simulations appears over other tropical oceans. While the oceanic tropical rainbelt is relatively well reproduced in IFS_F-C4 in terms of pattern, the oceanic tropical rainbelt is less well reproduced in ICON-C4. The westward extension of the tropical rainbelt is too small in ICON in the Atlantic Ocean, while IFS does extend westward similarly to observations. The meridional extent is, however, underestimated. In the Indo-Pacific region, ICON-C4 underestimates precipitation in the equatorial region, causing the well-known double ITCZ bias in this region, observed in traditional climate models. This bias is evident in the zonal mean precipitation over the western Pacific (Figure 5c). In the equatorial western Pacific, ICON-C4 simulates 7 mm d$^{-1}$ less precipitation than what is expected from observations. In the case of IFS_F-C4, the dry bias at the equator over the western Pacific is smaller than in ICON-C4. Precipitation is 3 mm d$^{-1}$ lower than IMERG in the equatorial western Pacific. The bias in IFS_F-C4 vanishes in the equatorial Pacific when IFS is coupled with the NEMO ocean model (Rackow et al., 2025b), pointing to different ocean surface representations and coupling choices as potential drivers of the development of this bias in IFS. IFS_F-C4 also presents a reduced area of tropical rainbelt in the Bay of Bengal and over Southeast Asia (Figure 5b).

Overall, precipitation over tropical oceans appears to be less constrained than over land. The remaining biases in precipitation over the tropical Pacific are linked to deviations in the tropical SST patterns, as observed in traditional climate models (Lin, 2007). Figure 6 shows the yearly-mean pattern of SST for the 2021-2025 period of simulation in IFS_F-C4 and ICON-C4, and the pattern of SST for the HadISST climatology (2001-2020). In the Pacific, the ICON-C4 and IFS_F-C4 represent the cold-warm gradient between the Eastern and western Pacific similar to observations. However, in ICON-C4, there is a westward extension of cold waters reaching the western Pacific, constraining the development of the West Pacific warm pool. In IFS_F-C4, this bias is not as prominent, with less westward extension of the cold tongue. In Figure 6, the warmest regions in the western Pacific and the western Atlantic are weaker than in observations. The West Pacific warm pool in IFS_F-C4 is 3-4 K warmer than the tropical SST mean, while in observations, the warm pool can be 4-5 K warmer than the tropical mean. The 1 K difference is also observed in the western Atlantic. In ICON-C4, the western Pacific warm pool is only 2 K warmer than the tropical mean. A similar value is observed in the western Atlantic.

The comparison between ICON-C4 and IFS_F-C4 raises some preliminary conclusions pointing to the positive and negative surprises in developing the two models. The fact that ICON-C4 can represent the tropical rainbelt over land and the Eastern Pacific indicates that a horizontal grid spacing of the order of 10 km is *sufficient* to reproduce the structure of precipitation in those regions, and that is possible with a minimum set of parameterizations. This is supported by the fact that across the

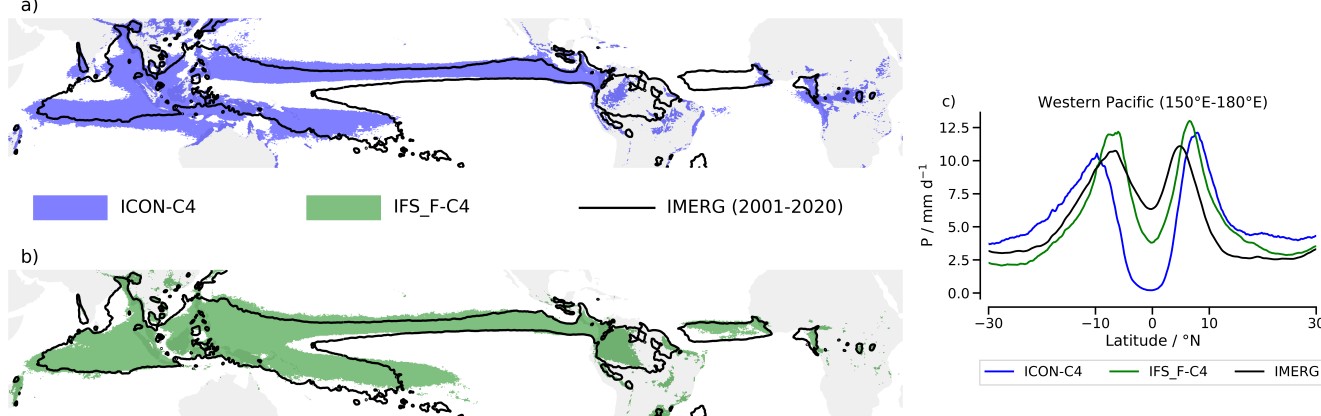

**Figure 5.** On the left, tropical rainbelt in ICON-C4 (a) and IFS_F-C4 (b) averaged over 2021 to 2025. The tropical rainbelt from IMERG (Huffman et al., 2019) averaged over 2001 to 2020 is outlined by black contour lines. On the right, zonal mean precipitation corresponding to the rainbelts on the left averaged over the western Pacific (c).

nextGEMS cycles, the structure of the tropical rainbelt presents negligible changes in the Eastern Pacific and land in ICON with horizontal grid spacing finer than 10 km. Indeed, the pattern of the tropical rainbelt over those regions is similar to the one presented by Segura et al. (2022).

On the other hand, using a grid spacing of 10 km is *not sufficient* to represent the tropical rainbelt in the western Pacific. To address this bias, IFS and ICON take different pathways, but both involve model tuning. IFS addresses this issue by using a convective parameterization. This is based on the long history of IFS in model tuning to match the observed precipitation pattern. ICON, using a simplistic framework regarding parametrizations, aims to address the warm pool precipitation bias by fine-tuning subgrid-scale processes (microphysics and turbulence). The results from Takasuka et al. (2024) and Segura et al. (2025a) show that getting a single tropical rainbelt in the western Pacific is possible with this simplistic framework. While Takasuka et al. (2024) and Segura et al. (2025a) used SST-prescribed simulations, the next step in the development of ICON is to include the changes proposed by these authors in coupled simulations. Moreover, the difference between ICON-C4 and IFS_F-C4 shows that a better representation of the equatorial SST pattern and the tropical rainbelt are linked, suggesting that once the tropical rainbelt is well reproduced, the SST pattern might follow.

### 4.3 Local- to synoptic-scale phenomena

In Section 4.1 we demonstrated that the nextGEMS simulations result in an energetically consistent global climate. As shown in Section 4.2, some of the typical spatial precipitation and SST patterns in the Pacific are acceptably reproduced, but some long-standing issues remain. In the following subsections, we investigate to what extent an energetically consistent climate translates into a constraint for local, meso-scale and synoptic-scale phenomena.

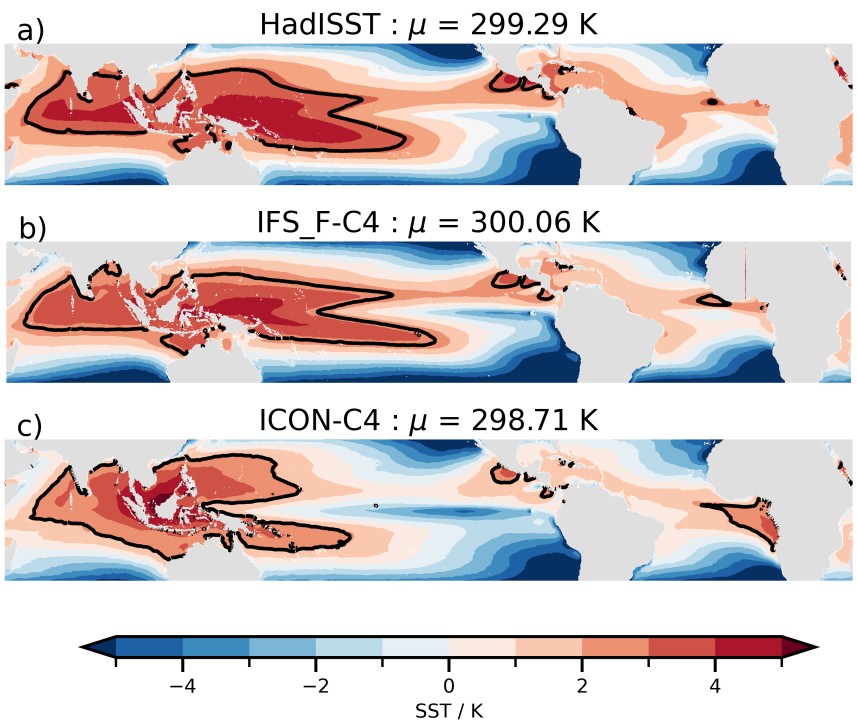

**Figure 6.** Annual mean sea surface temperature (SST) from HadISST (Rayner et al., 2003) averaged over 2001 to 2020 (a) and from IFS_F-C4 (b) and ICON-C4 (c) averaged over 2021 to 2025. The spatial mean, denoted as $\mu$ in the subplot titles, is subtracted from each product. And the 80th percentile is outlined by black contour lines.

### 4.3.1 Local-scale phenomena

We here take the soil moisture–precipitation feedback as an exemplifying process of how the two models represent a local-scale phenomenon. The reasoning behind this is that local convection, on scales below 100 km, plays a dominant role in this coupling. The soil moisture-precipitation feedback is already apparent in the first year (Figure 7). Both ICON-C4 and IFS_F-C4 simulations generally reproduce similar patterns of where strong and weak soil moisture–precipitation feedback occurs within the first five years. This is quantified by the correlation coefficient between total-column soil moisture and precipitation during the boreal summer. The two models agree with each other on the regions with relatively strong feedback, such as Mexico, the southern United States and the Sahel. These strong feedback regions align well with the hot spots of land-atmosphere coupling where evapotranspiration is strongly controlled by soil moisture (Koster et al., 2004). The models do not share all the features. ICON-C4 generally shows weaker feedback strength globally than IFS_F-C4. The reason for the weaker feedback can be attributed, in part, to how each model represents convection (Lee and Hohenegger, 2024). Convective parameterization causes precipitation to be more sensitive to surface evapotranspiration, leading to stronger feedback compared to the results with explicit convection (Hohenegger et al., 2009). The time series of land-atmosphere feedback over Europe, the Sahel, the

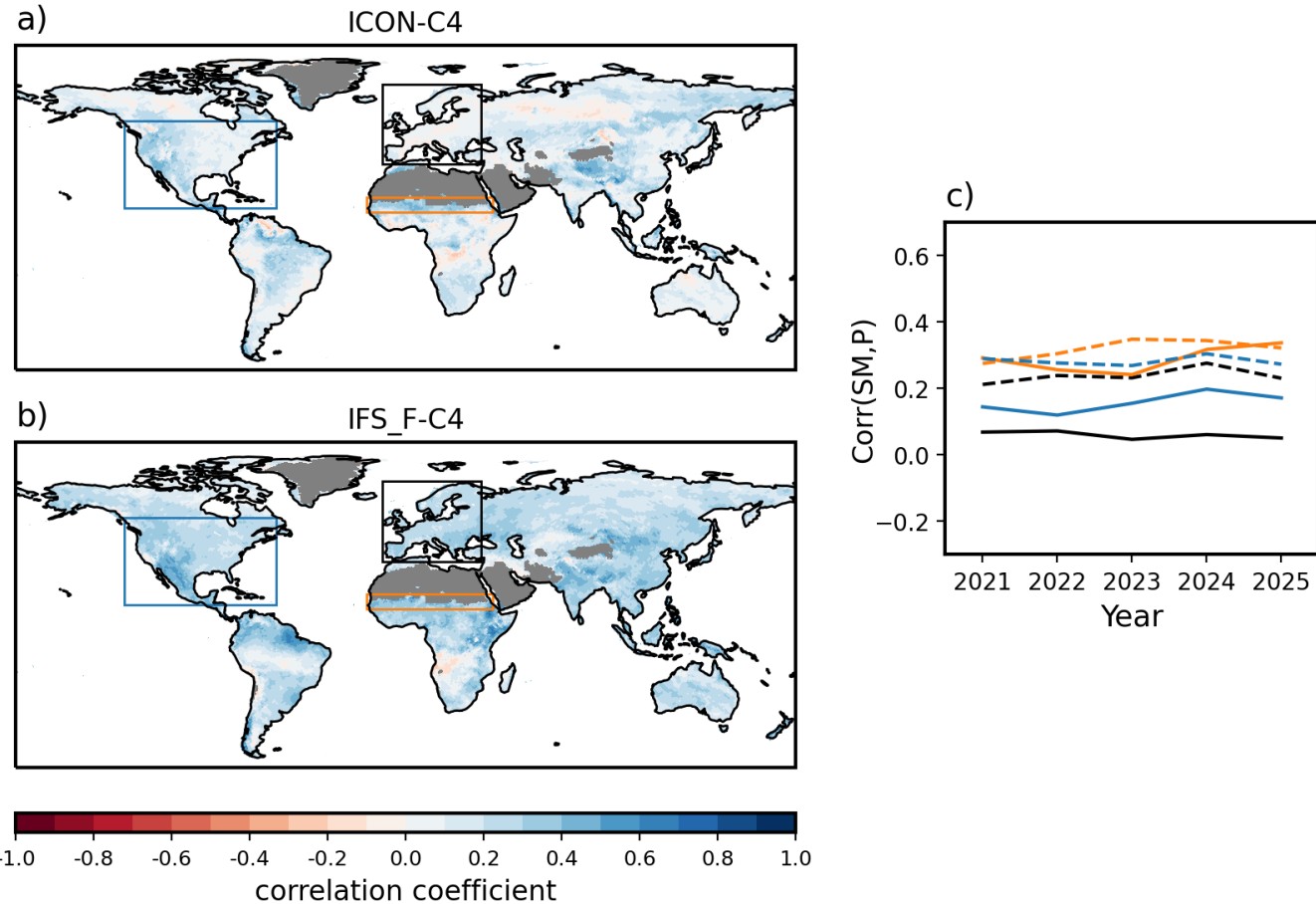

**Figure 7.** Soil moisture–precipitation feedback quantified with the correlation between daily mean total-column soil moisture (SM) and precipitation (P) during JJA of 2021 to 2025 for ICON-C4 (a) and IFS_F-C4 (b). Time series of the correlation coefficient (c) in ICON-C4 (full lines) and IFS_F-C4 (dashed lines) averaged over Europe [black box in (a) and (b)], Sahel [orange box in (a) and (b)], and the United States and Mexico [blue box in (a) and (b)]. Areas where the precipitation is smaller than $0.1\,\mathrm{mm\,d^{-1}}$ in both simulations are grayed out.

United States, and Mexico show that the correlation coefficients established in the first year remain relatively stable over time, with only minor fluctuations due to interannual variability.

### 4.3.2 Meso-scale phenomena

As an example of a meso-scale phenomenon, we look into maritime stratocumulus cloud fields, typically occurring over cool currents off the western side of continents.

Of relevance for the representation of stratocumulus clouds in nextGEMS models were the following specifications. First of all, we note that ICON-C4 refuses any parameterization of convective clouds, while IFS_F-C4 employs a shallow convection

scheme as well as a modified deep convection scheme. It was already discussed in Section 4.1 that for both ICON-C4 and IFS_F-C4 the *cloud inhomogeneity factor* ($\zeta$), which addresses the subgrid-scale inhomogeneities of clouds, was adjusted under the assumption that on the finer grids, greater inhomogeneity is explicitly represented. For IFS_F-C4 also a parameter controlling cloud edge erosion was reduced in order to increase cloud cover of low clouds. Both of these parameters affect cloud-radiation interaction and, as such, impact the radiation balance at the top-of-atmosphere, showing stratocumulus clouds as a good example of how microscale processes impact global scales.

In Figure 8a, we show the annual cycle of cloud water path for IFS_F-C4 and ICON-C4 and cloud area fraction for IFS_F-C4 and CERES satellite observations, averaged over the ocean near the coast of California, one of the main stratocumulus regions (Klein and Hartmann, 1993). Figure 8b shows albedo, given as the fraction of top-of-atmosphere outgoing shortwave radiative flux (RSUT) and incoming shortwave radiative flux, as well as RSUT individually. Note that for the simulations, we average over the years between 2021 and 2025, and for observations between 2000 and 2023, meaning that the periods hardly overlap. We recognize from the cloud properties and albedo two maxima in CERES observations, one in summer and one in winter. The annual cycle in albedo is well represented in ICON-C4, showing agreement in both summer and winter peaks. Disregarding the systematic differences of cloud water path, we find that particularly the summer peak is well represented in ICON-C4. In turn, IFS_F-C4 simulates a too-early peak in May and one in winter. Looking at RSUT, we find that the summer peak dominates the radiative effects of the Californian stratocumulus clouds. ICON-C4 simulates the annual cycle well, while IFS_F-C4 does not reach the summer peak amplitude in RSUT, which is in line with the absent summer peak in the cloud water path. The annual cycle of Californian stratocumulus presented here is well in line with the one discussed in a separate study by Nowak et al. (2025) on stratocumulus in km-scale Earth system models. These also authors examined the representation of shallow cumulus.

From the annual cycles shown in Figure 8, we identify summer (JJA) as the radiatively most relevant season of marine stratocumulus off the coast of California. In Figure 9 we show the associated spatial maps of RSUT for JJA along with the cloud properties and RSUT biases, here averaged over the 2021-2025 period for both simulations and the 2016-2020 period for CERES. For ICON-C4, but particularly for IFS_F-C4, RSUT is underestimated. In CERES, the stratocumulus cloud field off the coast of California, as shown in Figure 9c, can be seen in cloud fraction and relates clearly to RSUT. For ICON-C4 and IFS_F-C4, we find that clouds have too small a liquid water path or are too far off the coast. As a consequence, associated biases in RSUT are substantial.

We summarize that the nextGEMS simulations' energetically consistent climate comes along with a stratocumulus cloud field in the Bay of California whose radiative effects (RSUT) are well in phase with that of CERES satellite observations. In ICON-C4, we find both the bi-modality of the annual cycle of cloudiness and albedo, as well as the peak amplitude of RSUT, matching well with observations. For IFS_F-C4, just like in CMIP5 and CMIP6 (Jian et al., 2021), the peak amplitude of RSUT is underestimated, particularly in summer, when stratocumulus clouds have their strongest radiative effects. The reasons for this are the absence or ocean-wards dislocation of stratocumulus, potentially inaccurate cloud-radiative properties, and under-resolved mixing processes of stratocumulus. Eventually, it would be of great interest to investigate the implementation

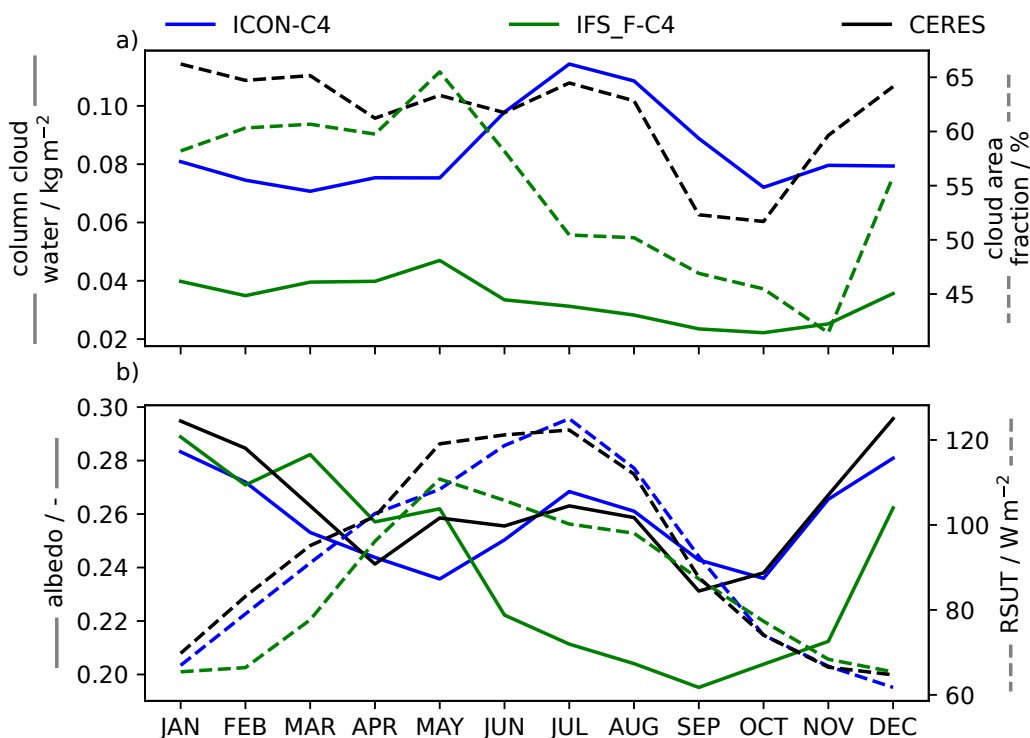

**Figure 8.** Annual cycle of stratocumulus cloud and radiative properties, averaged over the years 2021 to 2025 and over the ocean west of the Californian coast (longitude [115°W, 127.5°W] and latitude [27.5°N, 38°N], blue rectangle in Figure 9. Panel a) shows column cloud water (kg m$^{-2}$, solid lines) and cloud area fraction (%, dashed lines). Panel b) shows the the albedo (-, solid lines) and top-of-atmosphere upward radiative flux (RSUT / Wm$^{-2}$, dashed lines). Note that for ICON-C4 cloud fraction is not available, while for CERES column cloud water is not available.

of aerosol effects on cloud formation and brightening in km-scale Earth system models in regard to stratocumulus clouds and their radiative effects.

"Whether a realistic representation of global climate constrains patterns of regional phenomena?" can not be fully answered because a proof of causal relationship between the global climate and regional circulation is not feasible. Still, the representation of the exemplary regional circulation, stratocumulus cloud fields, is found satisfying. In particular, for ICON-C4, despite not employing a parameterization of shallow convection, we find that not only the phase but also the amplitude of stratocumulus cloud radiative effects agree very well with observations (Nowak et al., 2021). This is an encouraging result that demonstrates the capabilities of km-scale Earth system models, as well as that instead of using convective parameterization the fine-tuning of cloud microphysics is key to representing stratocumulus clouds.

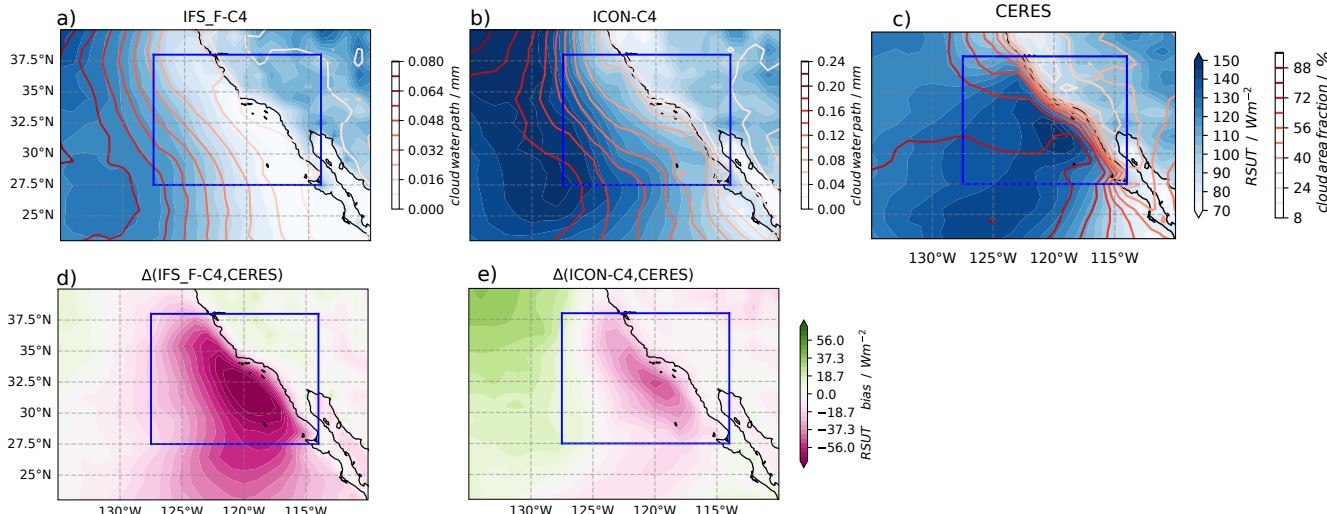

**Figure 9.** Top-of-atmosphere upwelling shortwave flux (RSUT) over the ocean next to the Californian coast in JJA for ICON-C4 (a), IFS_F-C4 (b) and CERES (Loeb et al., 2018) (c), along with the cloud (liquid) water path ($kg\,m^{-2}$) for nextGEMS Cycle 4 models [2021; 2025] and cloud area fraction (%) for CERES [2016; 2020]. Panels (d) and (e) show biases in RSUT of ICON-C4 and IFS_F-C4 relative to CERES, respectively. The blue rectangle denotes the area considered for computing RSUT statistics in Figure 8.

### 4.3.3 Synoptic-scale phenomena

Atmospheric blocking is a key feature of mid-latitude synoptic-scale circulation, often linked to weather extremes such as heatwaves and cold spells. Here, we use atmospheric blocking as an example of a synoptic phenomenon. Figures 10a,b illustrates the climatology of blocking events (Figures 10a,b) and the 2021–2025 time series of blocking events for the "North Pacific" and "Central Europe" regions (Figure 10c). Overall, there is good agreement in the geographical distribution of atmospheric blocking between ICON-C4 and IFS_F-C4. However, IFS_F-C4 exhibits a higher frequency of blocking events compared to ICON-C4.

We hypothesize that the convergence in blocking location results between these models is influenced by their shared utilization of kilometer-scale orographic information. As demonstrated by Davini et al. (2022) using coarser-resolution simulations, higher-resolution orography significantly enhances the representation of atmospheric blocking. Specifically, the improved performance of IFS-FESOM in capturing blocking frequency can be attributed to its implementation of turbulent orographic form drag and subgrid-scale orography parametrizations, both of which are known to enhance the representation of atmospheric circulation features (e.g., Woollings et al., 2018).

The time series of annual blocking frequencies for the North Pacific and Central Europe regions reveal substantial interannual variability in both IFS_F-C4 and ICON-C4. Note that recent studies indicate that there is no significant trend in blocking frequency over the past decades (Wazneh et al., 2021). Traditional climate models tend to underestimate atmospheric block-

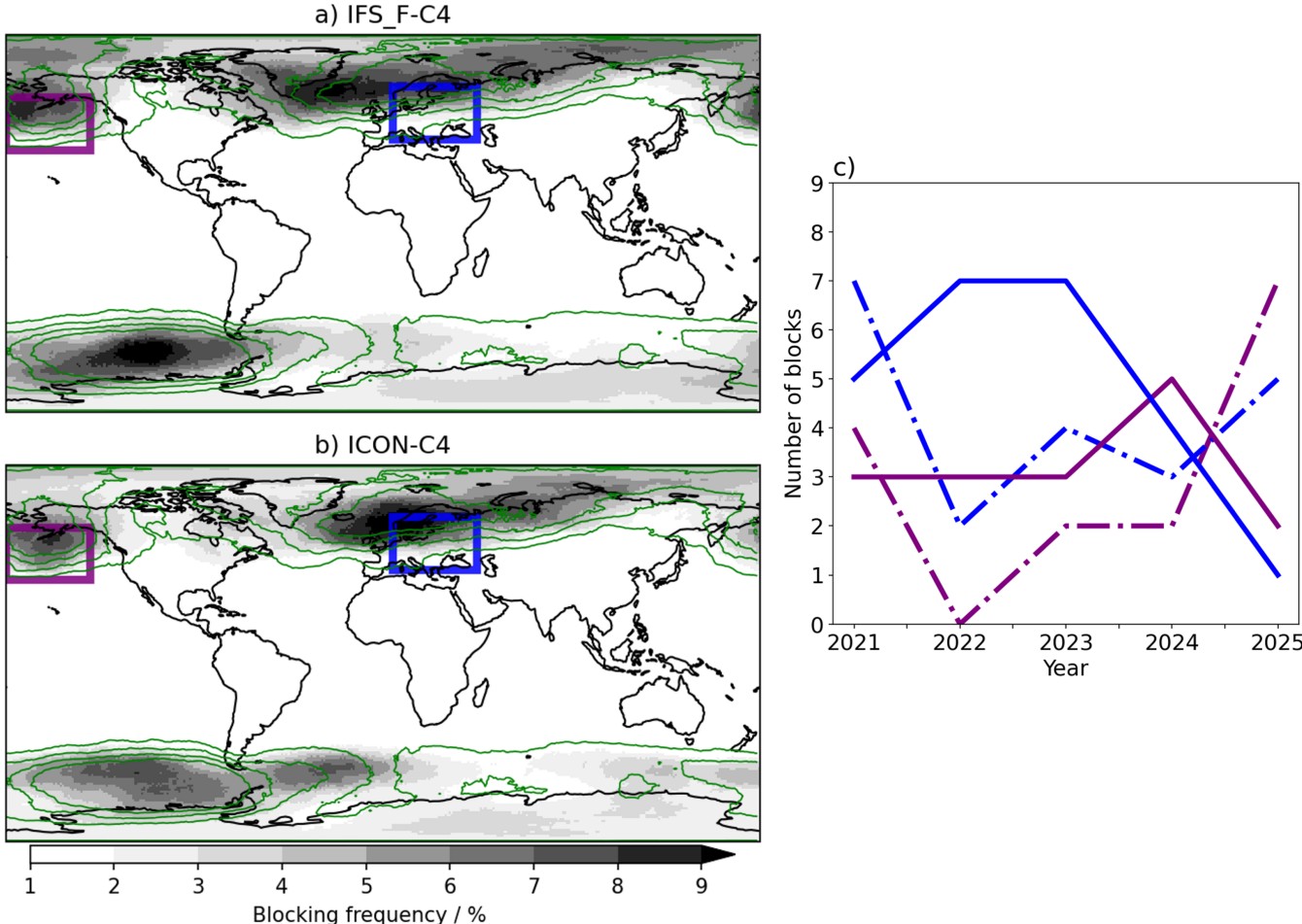

**Figure 10.** Annual blocking frequency in IFS_F-C4 (a) and ICON-C4 (d) during 2021–2025. Units are in percentage of blocked days relative to the total number of days per year. ERA5 climatology (2015–2019) is shown in green contours (2% intervals starting at 2%). Evolution of the number of blocks (c) in ICON-C4 (full lines) and IFS_F-C4 (dashed lines) passing over central Europe [blue box in (a) and (b)] and and North Pacific [purple box in (a) and (b)]. Blocking is identified as a persistent and quasi-stationary mid-level (500 hPa) geopotential height anomaly of the flow following the Schwierz index (Schwierz et al., 2004).

ing frequency (e.g., Dolores-Tesillos et al., 2025); thus, whether blocking will increase or decrease is still an open question (Berckmans et al., 2013; Davini and D'Andrea, 2016; Schiemann et al., 2017; Woollings et al., 2018).

## 4.4 Time scales of regional patterns

The analysis in Section 4.1 brought another positive surprise: The TOA energy balance responded quite quickly to changes in 525   the parameterization schemes. The tuning of the TOA energy balance shown in Section 4.1 was achieved in parallel in ICON

and IFS-FESOM by performing a number of short integrations. In the process, selected parameter values were modified within observational uncertainty to obtain a simulated TOA energy balance similar to the observed one. Indeed, the integration time was no longer than 12 days. This strategy was further used in IFS for testing across different resolutions. This means that with only a few days of km-scale simulations, one can build enough confidence on whether changes in parameter values converge to an improved response. Such a finding reinforces the possibilities of these km-scale models to be used more routinarily and in an operational context.

In the same line, robust regional circulation patterns in Section 4.2, Section 4.3.1 and Section 4.3.2 emerge within relatively short timespans, e.g. one-year periods. The double band of precipitation in the Western Pacific in ICON-C4 is present every year of the simulation and is already observed in the first 3 months of the simulation (not shown). The annual cycle of stratocumulus clouds is established already in the first year (Figure 8). Likewise, the soil moisture-precipitation feedback emerges already during the first year of simulations (Figure 7). The lesson that we want to communicate here is that it is unnecessary to run lengthy simulations to understand how parameterization changes will affect the TOA energy balance, the land-atmospheric coupling, the double ITCZ, and stratocumulus clouds. Thus, this lesson could be helpful for communities working on those problems. On the other hand, the large interannual variability on atmospheric blockings both in ICON and IFS-FESOM combined with the few occurrences of blocking in a given year (Fig. 10) suggests that multidecadal simulations are required to assess the performance and evolution of blocking frequencies properly.

## 5 Conclusions

In 2021, nextGEMS started with the goal of producing for the first time multidecadal climate simulations, in which the governing equations are solved at a horizontal resolution on the order of 10 kilometers or finer in the ocean, land, and atmosphere. With the resources available, nextGEMS provided 30-year climate simulations (from 2020 to 2050) under the SSP3-7.0 scenario with two different Earth system models, ICON and IFS-FESOM, with a horizontal grid spacing of 10 km in the atmosphere and 5 km in the ocean. The 30-year climate simulations were run on the supercomputer Levante, using 18% of its capacity in the case of ICON and 9% in the case of IFS-FESOM. While the limited computing resources forced the final horizontal grid spacing to be 10 km, nextGEMS prepared ICON and IFS-FESOM to be run on more powerful supercomputers than Levante, exploiting computer resources to produce climate simulations with a horizontal grid spacing of 5 km. The achievement of the 30-year climate simulations inspired this overview. Here, we presented the concept and progress of the project structured into four cycles and hackathons. In addition, we discussed the surprises we encountered and the lessons we learned when developing the models and analyzing the simulations. To create a storyline, we translated our learning process into four questions defined in Section 1.

Over the four cycles, the simulations with ICON and IFS-FESOM evolved from year-long simulations in Cycle 1 with significant mass and energy leaks to multi-decadal simulations in Cycle 4 with an energetically consistent climate, i.e., the response of the surface temperature to the radiative forcing was comparable to observations. The two 30-year simulations of Cycle 4, with ICON and IFS-FESOM, had a horizontal grid spacing of about $10\,\mathrm{km}$ in the atmosphere and land and $5\,\mathrm{km}$ in

the ocean. Notably, both simulations had a competitive throughput of 414 simulated days per day on 464 nodes for ICON and 600 simulated days per day on 269 nodes for IFS-FESOM.

In both models, ICON and IFS-FESOM, it was relatively easy to tune the cloud properties and bring the TOA radiation balance close to observations. The main reason was the quick response of the TOA radiation balance to changes in the cloud properties, allowing us to find optimal parameters with short simulations of about 12 days. The cloud properties were adjusted in different ways in ICON and IFS-FESOM. While ICON targeted shallow and low-level stratocumulus clouds, IFS-FESOM targeted all types of clouds. Despite these different approaches, the bias in the TOA radiation balance could be reduced in both models, with a closer fit in IFS-FESOM.

The comparison between ICON and IFS-FESOM also gave insights into the representation of the mean climate. A km-scale model with no convection parameterization like ICON can reproduce the observed structure of the tropical rainbelt over land in the Eastern Pacific, similar to IFS-FESOM, which uses a convection parameterization. However, ICON can not reproduce the pattern of the tropical rainbelt (still showing a double ITCZ) or the SST patterns in the western Pacific. Here, IFS-FESOM behaves more similarly to observations. Thus, solving explicitly km-scale processes with a grid spacing of 10 km is not sufficient to solve biases in the western Pacific, but fine-tuning in km-scale Earth system models can reduce the double ITCZ bias (Takasuka et al., 2024).

We also observed in ICON and IFS-FESOM that the seasonal cycle of stratocumulus clouds, and pattern of atmospheric blocking are well represented, i.e., similar to observations. This means that these regional phenomena are well constrained in km-scale climate simulations. However, there are some differences. ICON reproduces the seasonal cycle of the stratocumulus clouds better than IFS-FESOM. The different pathways for fine-tuning radiative properties of low-level clouds explain the discrepancy between ICON and IFS-FESOM. We also observe that the use of convective parameterization is not necessary to represent this regional phenomenon correctly. On the other hand, the frequency of atmospheric blocking occurrence is better represented by IFS-FESOM, signaling the importance of sub-scale processes to represent the frequency of this synoptic circulation.

Our analysis also showed that short simulations of one year or even less are sufficient to develop and test many aspects of km-scale models such as biases in the tropical rainbelt, in the top-of-atmosphere radiation balance, the seasonal cycle of stratocumulus clouds, or the soil moisture–precipitation feedback. This finding is particularly encouraging for other groups who work on resolving these biases or understanding the changes in regional circulation patterns under different climate scenarios. The resources for such short simulations are available on any of the EuroHPC systems, which brings us one step closer to the vision of using km-scale models in a research and operational setting.

## 6   Outlook

The development of ICON and IFS-FESOM continues within and beyond nextGEMS. To further increase the resolution and throughput, modeling centers are increasingly making use of exascale supercomputers such as the pan-European Large Unified Modern Infrastructure (LUMI, 2024). ICON, and to a lesser extent IFS-FESOM, have been adapted to perform well on such

machines primarily based on graphical processing units (Adamidis et al., 2024; Bishnoi et al., 2024). nextGEMS collaborates with other projects, such as WarmWorld (https://warmworld.de), on the optimization and modularization of the code base of ICON, and with EERIE (https://eerie-project.eu), on the role of mesoscale processes in the ocean.

To improve the representation of the Earth system, ICON is currently including more components, such as the HAMOCC module for ocean biogeochemistry (Ilyina et al., 2013) and the HAM-lite module for interactive aerosols (Weiss et al., 2025). HAMOCC and HAM-lite have been evaluated with complementary simulations of up to one year. In preliminary simulations with ICON-HAMOCC at 10 kilometer resolution, biochemistry patterns at the regional scale appeared well reproduced in the case of West African waters. On the other hand, sea surface chlorophyll-a begins to be reconstructed (Roussillon et al., 2023), which can be used to forecast marine productivity and its effect on shifts of exploited fish populations (Sarre et al., 2024) as well as trends in pelagic biomass (Diogoul et al., 2021). In simulations with ICON-HAM-lite at 5 kilometer resolution, key aerosol processes are captured including, for example, the formation of dust storms in the Sahara, wind-driven emissions of sea salt aerosols, or wildfire-driven emissions of carbonaceous aerosols. Figure 11 shows a scene of aerosol burdens on 5 September 2020 at 00:00 UTC. These examples show how the simulations produced so far, and those to come can be of interest for industry sectors such as solar and wind energy or agriculture and fisheries.

The two models developed in nextGEMS are also prototypes for the Climate Change Adaptation Digital Twin of the Destination Earth initiative (https://destination-earth.eu). The initiative aims to operationalize km-scale and multi-decadal climate simulations, assess the impacts of climate change, and evaluate adaptation strategies at local and regional scales (Hoffmann et al., 2023; Sandu, 2024). For that initiative, the first projections from 2020 to 2040 were produced with ICON, IFS-NEMO, and IFS-FESOM at resolutions of $5\,km$ and $4.4\,km$ in the atmosphere and $5 - 10\,km$ in the ocean, respectively. Moreover, the work on output harmonization in nextGEMS fed back into Destination Earth. The output of the models underpinning the Digital Twin is converted into a Generic State Vector, a set of selected variables in consistent units, frequencies, and on a common grid, which is then streamed to the different applications. nextGEMS also participated in the Global KM-scale Hackathon of the World Climate Research Programme by providing several simulations with ICON and IFS-FESOM, including some at 2.5 km resolution over 14 months. These simulations are publicly available on GitHub (https://digital-earths-global-hackathon.github.io/catalog/) together with those of many other participating models.

nextGEMS is the first step towards a European ecosystem of km-scale climate research and operational Earth system models, to which WarmWorld, EERIE, Destination Earth, and other projects contribute. These projects embody a cross-European effort to push forward the boundaries of climate science by reducing model uncertainties and providing climate information on local and regional scales where the impacts of climate change are felt (Prein et al., 2015; Gettelman et al., 2023). In this context, the Earth Virtualization Engine initiative (https://eve4climate.org) understood the necessity to provide fair free access to climate information with local granularity, globally, using the best technology at hand (Stevens et al., 2024). Thus, the km-scale Earth system models developed in nextGEMS will not only serve scientific purposes but will also be tools to address critical questions for society and ecosystems regarding climate adaptation, mitigation, and risk management.

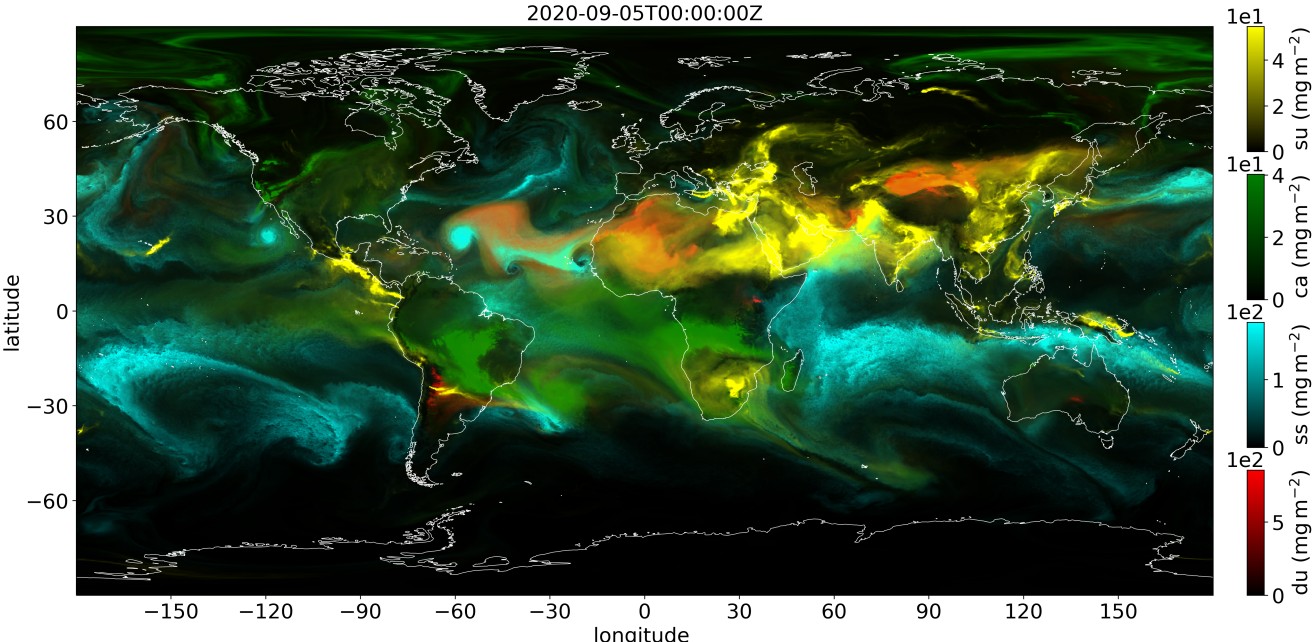

**Figure 11.** Scene of aerosols on 5 September 2020 at 00:00 UTC. Shown are column burdens of dust (du) in red, sea salt (ss) in blue, carbonaceous aerosol (ca) in green, and sulfuric aerosol (su) in yellow. The colormaps have a variable transparency which decreases from fully transparent at minima to fully opaque at maxima.

*Code availability.* The ICON model is available on the WDCC (https://doi.org/10.35089/WDCC/IconRelease01, ICON partnership (DWD, MPI-M, DKRZ, KIT, C2SM), 2024) under a permissive open source licence (https://opensource.org/license/BSD-3-clause). The FESOM2.5 model is a free software and available on Github (https://github.com/FESOM/fesom2). The latest version 2.5 including all developments used in nextGEMS Cycle 3 is archived on Zenodo (https://doi.org/10.5281/zenodo.10225420, Rackow et al., 2023b). MultIO is a free software and available on the GitHub of ECMWF (https://github.com/ecmwf). The IFS model is available subject to a licence agreement with the ECMWF. ECMWF member-state weather services and approved partners will be granted access. The IFS code without modules for data assimilation is available for educational and academic purposes via an OpenIFS licence (http://www.ecmwf.int/en/research/projects/openifs). The IFS code modifications for nextGEMS are archived separately on Zenodo (https://doi.org/10.5281/zenodo.10223576, Rackow et al., 2023a). The data and scripts that we used to generate the figures are available on the the Open Research Data Repository of the Max Planck Society (https://doi.org/10.17617/3.QZHXMC, Segura et al., 2025b).

*Data availability.* The simulation data is openly accessible and archived on the WDCC: Cycle 2 (https://doi.org/10.26050/WDCC/nextGEMS_cyc2, Wieners et al., 2023), Cycle 3 (https://doi.org/10.26050/WDCC/nextGEMS_cyc3, Koldunov et al., 2023), and Cycle 4 (https://doi.org/10.35095/WDCC/nextGEMS_prod_addinfov1, Wieners et al., 2024). Namelist files and settings for the 30-year Cycle 4 production simulation with IFS-FESOM are archived on Zenodo (https://doi.org/10.5281/zenodo.14725225, Rackow et al., 2025a). The IMERG data

## Appendix A: Summary of model developments

Here, we summarize all developments in ICON over cycle 2 to 4.

### A1 Cycle 2

In the atmosphere module, several bugs in the dynamical core, cloud microphysics, and surface fluxes were removed, which reduced the energy imbalance significantly. The old PSrad radiation scheme was replaced with the new RTE-RRTMGP scheme (Pincus et al., 2019). A cloud inhomogeneity factor was introduced to tune the radiation balance at the top of the atmosphere. In the land module, the GSWP3 input data of 2014 (Dirmeyer et al., 2006) was used to spin up the land reservoir for the hydrological discharge. The equilibrium was reached after 5 years using a time step of $40\,\mathrm{s}$. The distribution area of freshwater entering the ocean was increased to a radius of $30\,\mathrm{km}$. In the ocean module, a new vertical coordinate system with thinner surface levels was introduced, and a new surface flux scheme was implemented. Moreover, the spin-up procedure was revised. The spin-up simulation was initialized with ORAS5 (Copernicus Climate Change Service, 2025) and forced with ERA5 (Hersbach et al., 2020) from 2010 to 2020 to establish a stationary eddy field. The sea surface temperature was nudged to ERA5 in the last year. After 8 test runs, two simulations were provided. ICON-C2-A used a horizontal grid spacing of 5 km and was integrated for 2 years, whereas ICON-C2-B used a horizontal grid spacing of 10 km and was integrated for 10 years. ICON-C2-A and ICON-C2-B were both performed on on 400 nodes. The throughput was 80 SDPD for ICON-C2-A and 550 SDPD for ICON-C2-B. Coarsening the resolution by a factor of 2 increased the throughput by a factor of 7.

### A2 Cycle 3

In the atmosphere module, the energy imbalance was further reduced by removing bugs in the cloud microphysics and turbulence scheme. The surface fluxes were calculated with the heat capacity at constant pressure instead of constant volume, which reduced the amount of energy transferred from the surface to the atmosphere by about $29\,\%$. In the land module, the heat capacity and conductivity maps were revised and the soil texture was accounted for in the hydrology scheme. In the ocean module, the vertical mixing was solved including Langmuir turbulence in the turbulent kinetic energy scheme. The turbulent diffusion coefficient was decreased from 0.2 to 0.1. The tracer advection was treated with a new second-order scheme. The spin-up of the ocean was conducted in the same manner as in Cycle 2 except for the year 2019 being repeated. The dynamical core was solved with mixed precision, and the sea level pressure was coupled to the atmosphere. In contrast to the previous cycles, the output was compiled on the HEALpix grid (Górski et al., 2005). After 27 test runs, ICON-C3 was integrated over 5 years using a horizontal grid spacing of 5 km in the ocean and atmosphere. The throughput of ICON-C3 was 98 SDPD on 530 nodes compared to 80 SDPD on 400 nodes in ICON-C2-A.

## A3  Cycle 4

In the atmosphere module, the diffusivity in the conservation of momentum was modified to account for the persistent leak of internal energy. The turbulent mixing scheme VDIFF was updated and renamed to TMX. This update came with several key improvements: refactoring the code into an object-oriented structure, enabling both explicit and implicit numerical solvers, replacing the implicit atmosphere-surface coupling with an explicit coupling, and replacing the dry static energy with the internal energy in the diffusion scheme. Lastly, the cloud inhomogeneity factor was linked with the lower tropospheric stability to better account for the cloud type. In the ocean module, the number of vertical levels was decreased from 128 to 72 levels, while keeping thin layers in the upper ocean, and the turbulent mixing under sea ice was reduced. The ocean state spin-up was conducted in the same manner as in Cycle 2 but without nudging of the sea surface temperature. ICON-C4 used ozone and greenhouse gas concentrations following CMIP6 for the forcing time series of a SSP3-7.0 scenario (O'Neill et al., 2016). In addition, ICON-C4 used the time-varying MACv2-SP climatology (Stevens et al., 2017) for anthropogenic aerosols together with the Stenchikov climatology from 1999 for volcanic aerosols (Stenchikov et al., 1998). After numerous long test runs, ICON-C4 was integrated over 30 years using a horizontal grid spacing of 10 km in the atmosphere and 5 km in the ocean. The 10 km grid spacing of the atmosphere, allowed us to reach a throughput of 414 SDPD on 464 nodes.

*Author contributions.*  HS, XP, PW, SM, TR, JL, ED, and IB analyzed the data, prepared the figures, and drafted the paper. MA, RA, GA, AB, JB, SB, EB, TB, SB, HB, NB, LB, SC, SD, JD, ID, PD, ME, JE, ME, RF, CF, LF, DG, PG, PG, AG, KG, MG, OG, HH, IH, KH, SH, JH, LK, AK, NK, TK, SK, SK, JK, PK, AK, RL, NM, MM, SM, KM, BN, JN, DP, UP, DP, RR, DS, DS, RS, PS, DS, DS, BS, DT, AT, AU, MV, MV, AV, SW, FW, MW, NW, KW, JW, MW, YW, FZ, and JZ contributed to the project and revised the paper. FB, DB, SB, SB, SB, PB, MD, ED, SF, EF, CH, CH, HJ, MJ, TJ, JJ, NK, DK, HK, MK, SM, OM, TM, JM, TM, EM, HP, KP, AS, PS, PS, LT, PV, IS, and BS conceptualized and contributed to the project and revised the paper.

*Competing interests.*  Chiel van Heerwaarden is a member of the editorial board of the Geoscientific Model Development (GMD) journal.

*Acknowledgements.*  This research was supported by the Horizon 2020 project nextGEMS under grant agreement number 101003470. Most simulations were performed and analyzed on facilities of the DKRZ (HLRE-4 Levante, 2024). We would like to thank DKRZ staff for their continued support in running the simulations and hosting and handling the data, in particular Jan Frederik Engels, Hendryk Bockelmann, Fabian Wachsmann, Irina Fast, and Carsten Beyer. We also want to thank the two anonymous reviewers for their insightful comments.

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
