# Peer review of "nextGEMS: entering the era of kilometer-scale Earth system modeling"

_EGUsphere, 2025_

## Author Response (AR1)

**Response to reviewers**

We thank both reviewers for the work and time invested in their careful review. The overall very positive evaluation of our manuscript made us satisfied with the direction and message that the manuscript aims to convey to the scientific community. The reviewers have identified a few minor shortcomings that require clarification. We address these shortcomings point-by-point in the following paragraphs.

**Reviewer 1**

This paper describes a major project involving many groups and many scientists to work towards high resolution climate modelling. The title is perhaps a bit tentative with the reference to "kilometre scale modelling", as the reported modelling is still an order of magnitude away from the kilometre scale.

The paper is very interesting from the organisational as well as the scientific point of view. It is a clear example of what can be achieved by bundling the expertise of many research groups and different types of expertise. Not only climate scientists and model developers are involved but also computer experts, software engineers, and application oriented people. Also the organisation of such a large number of people is worth reporting. Goals are set for each stage and hackathons are organised to discuss the results. It turns out to be an effective hands-on approach.

The science is very interesting with two rather contrasting atmospheric models and two ocean models. The key theme is the role of high resolution. The ICON atmospheric model has a minimalistic parametrisation and leaves cloud generation and convection to the dynamics of the model. The IFS model has highly developed parametrisations and gradually reduces the convective parametrisation activity at very high resolution. The key question for the ocean model is: what benefit does explicit modelling of ocean eddies bring?

The paper is very well written and carefully crafted. Given the strong scientific and organisational messages, the paper is well worth publishing. I recommend publication in its present form. The authors might want to make a few changes related to the comments below.

We thank the reviewer for the comment and for the appreciation of the manuscript.

Regarding the comment about the title, the word "kilometer scale modelling" is now specified in the manuscript to refer to simulations or models using a grid spacing of 10 km or finer. We added this definition on lines 22–23 of the Introduction:

"Such models or simulations are referred to as "km-scale simulations" or "km-scale models" in this manuscript."

Moreover, the title "entering the era of kilometer-scale Earth System model" indicates that nextGEMS is only the first step towards kilometer-scale climate simulations. In this sense, the lessons learned in nextGEMS are used in other European projects to run climate simulations with a grid spacing of 2.5 km or finer. We strengthened this argument on lines 78–81 of the Introduction:

"While climate simulations with a horizontal grid spacing of 10 km have been analyzed for this manuscript, nextGEMS also performed simulations with a horizontal grid spacing of 2.8 and 5 km integrated over shorter time periods. The lessons learned in nextGEMS are transferred to other projects, which conduct climate simulations with horizontal grid spacings up to 1.25 km. In this sense, nextGEMS is only the first step in a new era of km-scale climate simulations."

**and on lines 613-616 of the Outlook:**

"nextGEMS also participated in the Global KM-scale Hackathon of the World Climate Research Programme by providing several simulations with ICON and IFS-FESOM, including some at 2.5 km resolution over 14 months. These simulations are publicly available on GitHub (\url{https://digital-earths-global-hackathon.github.io/catalog/}) together with those of many other participating models."

**A few minor comments:**

Line 252-254

Interesting that it takes a comparison with another model to find a mistake in the use of c\_p and c\_v in the formulation of surface fluxes!

The use of c\_p instead of c\_v was among the possible theories for the energy leak in ICON. While this finding was not a direct outcome of a model comparison, the search in Cycle 2 was motivated by the fact that, unlike ICON, IFS did not show an energy leak.

**Section 4.3.1**

The discussion on soil moisture/precipitation feedback is very interesting and important. It is very undesirable to have positive feedbacks in a climate model that do not exist in nature. Such an erroneous feedback inevitably leads to the simulation of unrealistic extremes. A positive feedback is present in many large scale models with parametrised

convection and the current paper (and previous papers) suggests that the sign of the feedback is reversed in convection resolving simulations.

The question is whether it is a parametrisation issue or a resolution issue? The current paper is not conclusive. IFS has some parametrisation of convection also at high resolution, but its "effective resolution" may be less than in ICON.

There is evidence that a convective parameterization can induce a positive soil moisture-precipitation feedback (Hohenegger et al., 2009). Consistent with this, our ICON Cycle 1 simulations show a negative feedback when the convective parameterization is not used, supporting the hypothesis by Lee and Hohenegger (2024). Based on this reasoning, it is plausible that the positive feedback observed in IFS arises from the use of a convective parameterization. However, further analysis is required to confirm this. Additionally, the extent to which the soil moisture-precipitation feedback depends on model resolution remains largely unclear — for instance, it is not yet known at what resolution the negative feedback in ICON might shift to a positive one, assuming such a transition exists. The question raised by the reviewer is, though, of definite interest and should be examined in further studies.

My question is about the water budget. In summer, the main moisture source over large continental areas (e.g USA east of the Rocky Mountains) comes from land evaporation. This suggests strong re-cycling. The soil shows spring/summer drying but there is also runoff. At high resolution with intermittent convection there is a negative feedback where dry areas are preferred to trigger convection (e.g. Taylor et al.). How this works out at larger scales is not clear. Somehow it must change the mean water budget. If convection at small scales has a preference for dry areas, it means that the dry soil can absorb more water. Does it imply that averaged over large areas there is less runoff? In other words, if less water is taken out of the system then the water is still available for the re-cycling process? Did you look at runoff in these simulations?

Following the reviewer's suggestion, we compared the ratio of global mean surface runoff to precipitation in ICON-C4 and IFS\_F-C4 for the period 2021–2025 (Fig. R1). Our analysis shows that ICON produces stronger surface runoff than IFS, while the total runoff (surface + subsurface) remains relatively comparable between the two models. While land-atmosphere coupling likely contributes to this difference, other factors, such as differences in the land surface models, precipitation characteristics, or the initial soil moisture, play a significant role as well. It seems that the land surface model in ICON prefers stronger surface runoff, whereas the land surface model in IFS tends to generate more subsurface drainage. These results merit, by themselves, a deeper analysis, which could be done in a dedicated study.

Figure R1. Ratio of global mean surface runoff and precipitation (a) and ratio of global mean total runoff and precipitation (b) in ICON-C4 (blue) and IFS\_F-C4 (green) over 2021–2025.

**Fig. 8**

The legend is very confusing, and it takes some time to understand the logic: solid lines refer to the left scale and dashed lines to the right hand scale, colours refer to ICO-C4, IFS\_F-C4 and CERES. CERES does not have data in the top figure whereas it is in the top legend. Perhaps it is better to replace the legends by titles in which the logic is described.

We improved the clarity of the figure by showing the line styles of each variable along the y-axis labels:

**And we adapted the caption as follows:**

"Figure 8. Annual cycle of stratocumulus cloud and radiative properties averaged over the years 2021 to 2025 and over the ocean west of the Californian coast (longitude [115° W, 127.5° W] and latitude [27.5° N, 38° N], blue rectangle in Figure 9). Panel a) shows the cloud liquid water path (kg m-2, **solid lines**) and cloud area fraction (%, **dashed lines**). Panel b) shows the albedo (-, **solid lines**) and top-of-atmosphere upward radiative flux (RSUT / Wm-2, **dashed lines**). Note that for ICON-C4 cloud fraction is not available, while for CERES column cloud water is not available."

**Section 4.3.2**

The discussion about stratocumulus is very interesting. ICON does much better than IFS in the area considered and with tuning of the global radiation budget. However, the lack of contrast between the cumulus and stratocumulus regime seems to be shared by the two models.

In Section 4.3.2, we only intend to discuss stratocumulus clouds. From this, we can not detect a lack of contrast between cumulus and stratocumulus clouds. However, in

Nowak et al. (2025), the cumulus and stratocumulus regimes were examined using the Cycle 3 simulations. They found the distinct differences between the cumulus and stratocumulus regimes as shown, for example, in their Fig. 3. For example, the relationship between the top of atmosphere albedo and sea surface temperature (or inversion height and vertical wind speed at 700 hPa) indicates significant differences.

While our study does not look into the cumulus regime, we refer to Nowak et al. (2025) on lines 478–481 of section 4.3.2:

"The annual cycle of Californian stratocumulus presented here is well in line with the one discussed in a separate study by Nowak et al. (2025) on stratocumulus in km-scale Earth system models. These authors also examined the representation of shallow cumulus."

Is it correct to say that ICON is better in the strato-cumulus regime, and worse in the cumulus regime?

Our results show that ICON has a better representation in the stratocumulus regime. In this sense, the reviewer is right. Regarding the cumulus regime, we did not check into the representation of shallow cumulus, but Novwak et al. (2025) showed comparable skills in ICON and IFS for the shallow cumulus regime. If the reviewer refers to the cumulus convective regime, we follow the reviewer's reasoning. IFS-FESOM shows a better representation than ICON. However, it is still hard to infer if there is a connection between having a good representation of stratocumulus and a bad representation of the cumulus regime, or that one is a consequence of the other. So, while we agree with the reviewer's interpretation, we will not address this in the manuscript as a causal relationship can not be identified.

**Reviewer 2**

This is an interesting paper that gives a high level overview of the next GEMS project. It describes a large effort to debug, run and tune km-scale coupled climate model. It's one of the first efforts of its kind at this resolution. The work is broken into 4 cycles, each analyzing progressively more mature versions of the coupled model. The final cycle, with the most mature version of the model was then used to address, with partial success, four climate science questions. The authors looked at several well chosen aspects of the simulations in Seciton 4. The results in Section 4.2 are quite interesting, and I have some minor comments on that below. I appreciated the nice discussion and encouraging results for stratocumulus in Section 4.3.

We thank the reviewer for the comments and the appreciation for the results and discussion in Section 4 of the manuscript.

I only have minor comments:

1. Some of the issues related to the energy budget that were resolved during the early cycles appears to be coding errors and bugs. Would it have been more efficient to also have a low resolution version of the coupled model to detect and fix these issues? Although both these models are not designed to run at typical CMIP style resolutions (~100km), would such a configuration run and be sufficiently Earth-like to be appropriate for addressing some software and numerical issues?

The suggestion of the reviewer, indeed, applies to IFS, which was used to conduct test runs at a coarser resolution to resolve major bugs or errors. 28 km was the coarsest resolution used for scientific investigations, while 140 km was used for technical and software checks. While the current configuration of ICON does not make it possible to run it stably for long periods with a resolution coarser than 40 km, other configurations with coarser resolution did not show any sign of an energy leak in the atmosphere. In that sense, conducting simulations with finer resolution permitted the identification of bugs in both ICON and IFS, and this is another outcome of nextGEMS. Another reason for coarse simulations not being sufficient is the fact that the simulated top-of-atmosphere radiation budget is resolution-dependent, implying that running at coarser resolutions may not be that helpful in detecting model drifts or energy leaks.

On the other hand, proposed solutions were first tested at coarse resolutions before being implemented in the km-scale version of IFS. In ICON, proposed solutions were directly tested at fine resolutions of 10 km. Coarse simulations with a horizontal grid spacing of 140 km were only used to resolve particular technical problems. In addition, simplified idealized simulations were used to isolate and debug those parts of the code that caused the energy and water leak.

We added a discussion to section 3.1 on lines 203–211:

"Bugs related to the energy and water imbalance in ICON and IFS were already present in previous simulations at coarser horizontal resolutions, and another configuration in the case of ICON, without causing any evident problems. In other words, the impacts of such bugs were negligible at coarse spatiotemporal scales and only became evident at the much finer spatiotemporal scales simulated in nextGEMS – an important lesson learned. A more detailed discussion on the identification and resolution of bugs in nextGEMS was presented by Proske et al. (2024). In the case of IFS, the proposed solutions were first tested at coarser resolutions before being implemented in its km-scale version. The coarsest resolution for scientific tests was 28 km. In the case of ICON, the proposed solutions were directly tested at finer resolutions of 10 km. Only particular technical problems were addressed with tests at coarser resolutions of 140

km. In addition, simplified idealized cases were used to isolate and debug those parts of the code that caused the energy and water leak."

2. The paper makes extensive use of the "km-scale" adjective, and defines this as (for the atmosphere) "...using horizontal grid spacings equal to or less than 10km globally". This definition is a little optimistic, and I think most atmosphere model developers would not consider 10 km resolution "km-scale".

In the introduction, the authors then state that "Km-scale atmospheric simulations, also referred to as storm-resolving simulations, resolve deep convection, capturing mesoscale convective systems ..." Some of the references cited to support that Km-scale atmosphere (so here, that implies a 10km atmosphere) are convection resolving are Peters et al 2019, which is using a 2.5km and Becker et al 2021, which concludes "Our results suggest that deep convection is not completely resolved at a resolution of 9 or even 4 km"

Thus I found the introduction a little unclear in that it has claims that probably apply to km-scale models running a 2.5km or finer resolutions, and may not apply at 10 km resolutions. In particular, the implied claim that 10 km model can resolve deep convection and mesoscale convective systems is probably not correct.

The reviewer raises an important point: Which resolution allows us to fully resolve deep convection and km-scale processes? There is no easy answer to this question. In this manuscript, we refrained from discussing this question. However, we are sure that by using a horizontal grid spacing of 10 km or finer, km-scale processes in the atmosphere, land, and ocean are better represented – the finer the horizontal grid spacing, the better the representation of those processes. In that sense, when we refer to "km-scale", it is the use of a horizontal grid spacing equal or finer than 10 km. We make this point clear in the Introduction of the revised manuscript. Moreover, the nextGEMS project did not only focus on climate simulations with a horizontal grid spacing of 10 km but produced a hierarchy of climate simulations with a horizontal grid spacing of 10 km, 5 km, and 2.5 km. The simulations with a horizontal grid spacing of 2.5 km were integrated over a couple of years, but the goal was to simulate 30 years.

To clarify all these points, we replaced the word "resolved" with "represented" in lines 20, 23, 30, 38, 48, and 55 of the Introduction.

We clarify, on lines 22–23 of the Introduction, that we refer to "km-scale simulations" or "km-scale models" as those using a horizontal grid spacing of 10 km or finer:

"Such models or simulations are referred to as "km-scale simulations" or "km-scale models" in this manuscript.",

and on line 48:

"by using horizontal grid spacings of 10 km or finer",

and on line 55:

"representing km-scale processes with horizontal grid spacings of 10 km or finer".

We also clarify, on lines 78–81 of the Introduction, that nextGEMS produced not only on climate simulations using a horizontal grid spacing of 10 km but also climate simulations with finer resolutions:

"While climate simulations with a horizontal grid spacing of 10\,km have been analyzed for this manuscript, nextGEMS also performed simulations with a horizontal grid spacing of 2.8 and 5 km integrated over shorter time periods. The lessons learned in nextGEMS are transferred to other projects, which conduct climate simulations with horizontal grid spacings up to 1.25 km. In that sense, nextGEMS is only the first step in a new era of km-scale climate simulations."

3. Related to comment #2, in section 4.2, the authors claim, "The fact that ICON-C4 can represent the tropical rainbelt over land and the Eastern Pacific indicates that a horizontal grid spacing of the order of 10 km is sufficient to reproduce the structure of precipitation in those regions, and that is possible with a minimum set of parameterizations.", and, "The experiments conducted by Segura et al. (2024) and Takasuka et al. (2024) indicate that fine-tuning subgrid-scale processes can produce a correct representation of the tropical rainbelt without the use of convective parameterization."

I thought this claim was not well supported. The Segura and Takasuka references were more nuanced in their claims, and were from much more constrained prescribed-SST simulations. My superficial takeaway after reading section 4.2 was that one still needs convective parameterization (like in IFS) at this resolution. It would be good if the authors can strengthen their arguments for this result, especially for non-expert readers such as this reviewer.

We understand the point of the reviewer, and in the new version of the manuscript, we clearly state the message of this section. The message is that a horizontal grid spacing of 10 km allows us to identify which features of the tropical rainbelt are obtained

out-of-the-box without using a convective parameterization, and which are not and require model tuning.

Our results show that a horizontal grid spacing of 10 km is not sufficient to get the correct structure of the tropical rainbelt over the Indo-Pacific Ocean. Still, it is sufficient for the structure of the tropical rainbelt in the Eastern Pacific and over land. The latter is supported by the fact that across the nextGEMS cycles, the structure of the tropical rainbelt presents negligible changes in the Eastern Pacific and over land in ICON, using a horizontal grid spacing equal to or finer than 10 km, and is similar to the one presented by Segura et al. (2022).

We added a discussion on lines 423–426 of section 4.2:

"This is supported by the fact that across the nextGEMS cycles, the structure of the tropical rainbelt presents negligible changes in the Eastern Pacific and land in ICON with horizontal grid spacing finer than 10 km. Indeed, the pattern of the tropical rainbelt over those regions is similar to the one presented by Segura et al. (2022)."

Regarding the tropical rainbelt structure in the western Pacific, IFS and ICON take different pathways to address it, but both of them do so via tuning. IFS addresses this issue by using a convective parameterization. This is based on the long history of the weather forecasting version of IFS in tuning the model to match the observed precipitation pattern. ICON, using a simplistic framework regarding parameterizations, aims to address the warm pool precipitation bias by tuning the microphysics and turbulence schemes. The results from Takasuka et al. (2024) and Segura et al. (2025) show the possibility of getting a single tropical rainbelt in the western Pacific with this simplistic framework. Of course, this needs to be tested in a coupled experiment, and it is part of the following steps in ICON development.

To strengthen our argument regarding the structure of the tropical rainbelt in the western Pacific, we add a discussion on lines 427–436 of section 4.2:

"On the other hand, using a grid spacing of 10 km is *not sufficient* to represent the tropical rainbelt in the western Pacific. To address this bias, IFS and ICON take different pathways, but both involve model tuning. IFS addresses this issue by using a convective parameterization. This is based on the long history of IFS in model tuning to match the observed precipitation pattern. ICON, using a simplistic framework regarding parametrizations, aims to address the warm pool precipitation bias by fine-tuning subgrid-scale processes (microphysics and turbulence). The results from Takasuka et al. (2024) and Segura et al. (2025) show that getting a single tropical rainbelt in the western Pacific is possible with this simplistic framework. While Takasuka et al. (2024)

and Segura et al. (2025) used SST-prescribed simulations, the next step in the development of ICON is to include the changes proposed by these authors in coupled simulations. Moreover, the difference between ICON-C4 and IFS\_F-C4 shows that a better representation of the equatorial SST pattern and the tropical rainbelt are linked, suggesting that once the tropical rainbelt is well reproduced, the SST pattern might follow."

**References**

Nowak, J. L., Dragaud, I. C., Lee, J., Dziekan, P., Mellado, J. P., and Stevens, B.: A first look at the global climatology of low-level clouds in storm resolving models, Journal of Advances in Modeling Earth Systems, 17, e2024MS004340, 2025.

Proske, U., Brüggemann, N., Gärtner, J. P., Gutjahr, O., Haak, H., Putrasahan, D., and Wieners, K.-H.: A case for open communication of bugs in climate models, made with ICON version 2024.01, EGUsphere [preprint], 2024.

Yamazaki, D., Sato, T., Kanae, S., Hirabayashi, Y., and Bates, P. D.: Regional flood dynamics in a bifurcating mega delta simulated in a global river model, Geophysical Research Letters, 41, 3127–3135, 2014.

Yamazaki, D., Kanae, S., Kim, H., and Oki, T.: A physically based description of floodplain inundation dynamics in a global river routing model, Water Resources Research, 47, W04501, 2011.